# 🐱 KITTEN: A Knowledge-Integrated Evaluation of Image Generation on Visual Entities

**Hsin-Ping Huang**[1,2*]   **Xinyi Wang**[1*]   **Yonatan Bitton**[1]   **Hagai Taitelbaum**[1]
**Gaurav Singh Tomar**[1]   **Ming-Wei Chang**[1]   **Xuhui Jia**[1]   **Kelvin C.K. Chan**[1]   **Hexiang Hu**[1]
**Yu-Chuan Su**[1]   **Ming-Hsuan Yang**[1,2]

[1]**Google DeepMind**    [2]**University of California, Merced**

Reviewed on OpenReview: https://openreview.net/forum?id=wejaKS9Ps0

## Abstract

Recent advances in text-to-image generation have improved the quality of synthesized images, but evaluations mainly focus on aesthetics or alignment with text prompts. Thus, it remains unclear whether these models can accurately represent a wide variety of realistic visual entities. To bridge this gap, we propose KITTEN, a benchmark for **K**nowledge-**InT**egrated image genera**T**ion on real-world **EN**tities. Using KITTEN, we conduct a systematic study of recent text-to-image models, retrieval-augmented models, and unified understanding and generation models, focusing on their ability to generate real-world visual entities such as landmarks and animals. Analyses using carefully designed human evaluations, automatic metrics, and MLLMs as judges show that even advanced text-to-image and unified models fail to generate accurate visual details of entities. While retrieval-augmented models improve entity fidelity by incorporating reference images, they tend to over-rely on them and struggle to create novel configurations of the entities in creative text prompts. The dataset and evaluation code are publicly available at https://kitten-project.github.io.

## 1 Introduction

Recent advances in generative AI have revolutionized multimedia content creation. Large Language Models (LLMs) excel at knowledge-intensive tasks like question answering and summarization. Cutting-edge image generation models, such as Imagen (Saharia et al., 2022; Imagen 3 Team, 2024; Hu et al., 2024), DALL · E (Ramesh et al., 2021; 2022), and Stable Diffusion (Rombach et al., 2022), generate photorealistic and creative images from text. However, as these models become more capable and popular, assessing their reliability is crucial. Research on LLMs shows that even the most advanced models can generate inaccuracies, potentially undermining trust and causing societal harm (Muhlgay et al., 2023; Feng et al., 2023).

Despite increasing attention to factuality in LLMs, the accuracy of image generation models remains underexplored. Existing benchmarks mainly assess alignment with general text descriptions (Lin et al., 2015), compliance with image-editing instructions (Ku et al., 2024), or adherence to spatial relationships (Gokhale et al., 2022). However, they fall short in evaluating how well models generate images that faithfully reproduce the precise visual details of real-world entities, objects, and scenes. These should be grounded in trustworthy knowledge sources (see examples in Fig. 1). Recently, HEIM (Lee et al., 2024) introduces an evaluation suite for assessing various aspects of image generation, including the ability to generate entities such as historical figures or well-known subjects. However, real-world visual entities are far more diverse than those covered by HEIM, requiring a broader assessment. Moreover, HEIM primarily evaluates the alignment between generated images and entity names in text prompts. It fails to capture the fine-grained visual details essential for assessing the reproduction of visual-world knowledge, as nuances of real-world entities cannot be conveyed through text alone. Thus, it is essential to directly evaluate the fidelity of the generated images.

---

*Equal contribution

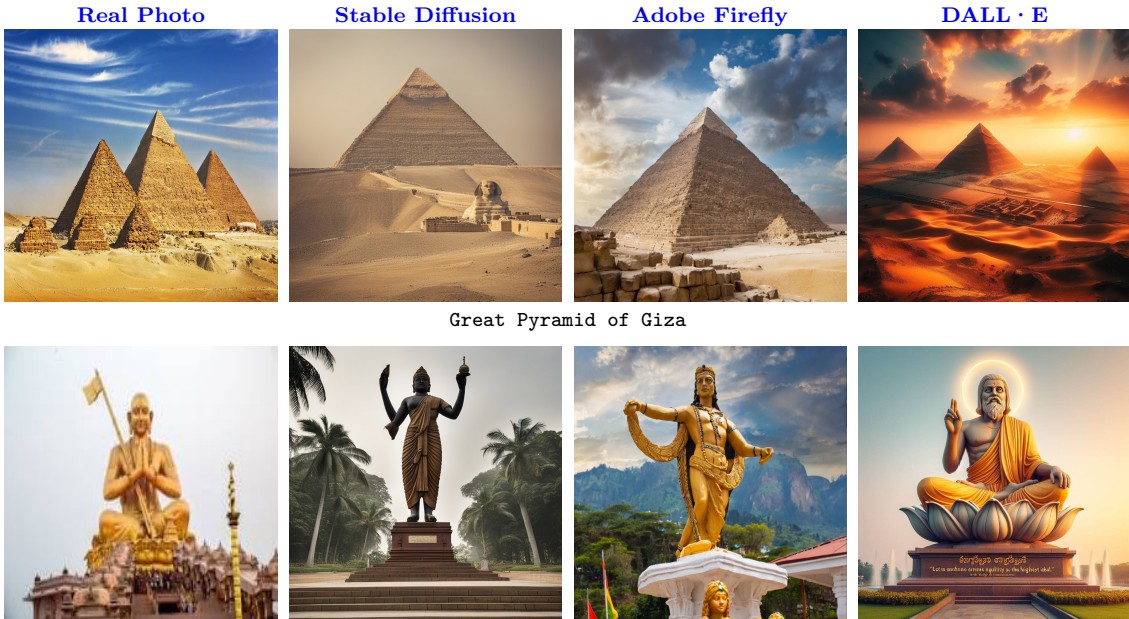

Figure 1: **Can text-to-image models generate precise visual details of real-world entities?** State-of-the-art models effectively render well-known entities (e.g., the Great Pyramid of Giza) but often struggle with less well-known entities, leading to hallucinated depictions.

To address the gap in evaluating image generation models' ability to reproduce visual-world knowledge, we introduce KITTEN, a benchmark dataset and evaluation suite designed to assess how well models generate visually accurate representations of real-world entities grounded in trustworthy knowledge sources. Unlike prior benchmarks that focus on aesthetics, text alignment, or commonsense reasoning, KITTEN uses prompts derived from visual entities documented in Wikipedia (Hu et al., 2023a), a reliable knowledge base, and evaluates real-world entities across eight visual domains (see Fig. 2). This ensures that generated images are compared with verifiable visual information crowdsourced from the internet. Additionally, we have developed a comprehensive set of human evaluation criteria that focus on the precise visual depiction of entities, capturing subtle but essential details for visual accuracy. By directly assessing entity fidelity in the generated images against established knowledge, KITTEN aims to advance the evaluation of world knowledge in image generation models.

Using KITTEN, we conduct a comprehensive evaluation of various text-to-image models, including standard, unified, and customization models fine-tuned or utilizing in-context learning with retrieved reference images (Chen et al., 2022). Our findings show that even the most advanced models (Imagen 3 Team, 2024; Black Forest Labs, 2024; Xie et al., 2024; Chen et al., 2025b) often fail to produce accurate representations, generating images that lack critical details essential for visual correctness. While retrieval-augmented models improve visual fidelity by incorporating reference images during testing, they tend to over-rely on these references, limiting their ability to generate novel configurations of entities from creative prompts. These findings highlight a key challenge in current image generation models: balancing entity fidelity with creative flexibility, underscoring the need for techniques that can generate precise visual details without sacrificing the ability to respond to diverse and imaginative user inputs.

## 2 Related Work

**Existing evaluation for text-to-image generation.** Evaluating text-to-image models has long been challenging, with many efforts aimed at improving performance measurement. Fréchet Inception Distance (FID) (Heusel et al., 2017) is a common metric for assessing perceptual quality by measuring the distribution gap between generated and real-world images. CLIP-T scores (Hessel et al., 2021) evaluate text-image

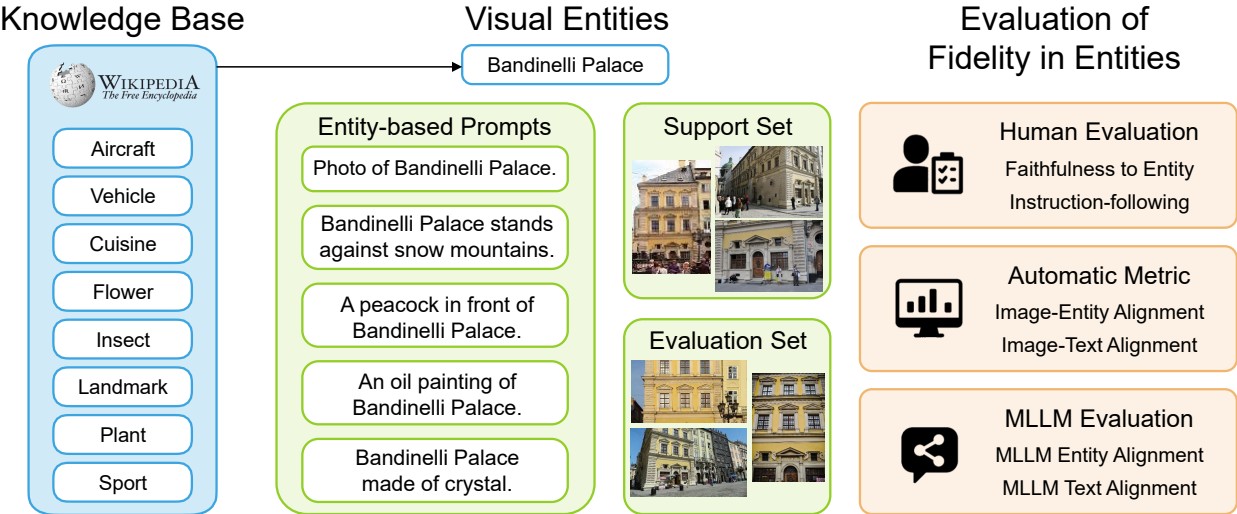

Figure 2: **KITTEN benchmark** is constructed from real-world entities across eight domains. For each selected entity, we define five evaluation tasks with image-generation prompts incorporating the entity. KITTEN includes a support set of entity images from the knowledge source for evaluating retrieval-augmented models, as well as an evaluation set for assessing the fidelity of the generated entities.

alignment by comparing the CLIP feature similarity between generated images and input prompts. These metrics summarize overall image quality. Several works assess alignment between generated images and text descriptions (Yarom et al., 2024; Gordon et al., 2023; Hu et al., 2023b; Wiles et al., 2024; Cho et al., 2023), but they mainly focus on semantic consistency rather than the fine-grained visual accuracy of depicted entities or specialized visual-world knowledge.

Recent works aim to evaluate models more thoroughly by decomposing the evaluation into subcategories such as attribute binding and numeracy, along with corresponding benchmarks. For example, the $SR_{2D}$ dataset (Gokhale et al., 2022) and the VISOR metric evaluate spatial relationships in text-to-image models, assessing whether objects in the generated image adhere to specified relationships (e.g., an orange *above* a giraffe). T2I-CompBench++ (Huang et al., 2023) contains text prompts from four categories (e.g., attribute binding) along with associated metrics. HRS-Bench (Bakr et al., 2023) evaluates model performance across five major groups (i.e., bias, fairness, generalization, accuracy, and robustness). TIFA v1.0 (Hu et al., 2023b) is a benchmark spanning 12 categories, paired with an automatic evaluation metric that measures image faithfulness via visual question answering. GenAI-Bench (Li et al., 2024) and ConceptMix (Wu et al., 2024) focus on evaluating compositional text prompts (e.g., objects with specific colors, shapes, or spatial relationships). Additionally, ImagenHub (Ku et al., 2024) evaluates models across different tasks by measuring semantic consistency and perceptual quality.

**Fidelity of entities in text-to-image generation.** While text-to-image models enable the creation of images from text descriptions, challenges arise when visual-world knowledge is required, i.e., generating accurate visual details of specific entities. Existing works have identified this issue and proposed solutions to mitigate hallucination (Lim & Shim, 2024). However, no clear methodology exists to systematically assess these models' limitations, which is crucial for further improvement. In this work, we propose KITTEN, a benchmark addressing the novel problem of evaluating image generation models' ability to produce fine-grained details of visual entities. We systematically assess the latest text-to-image, unified, and retrieval-augmented models using carefully designed human evaluations, automatic metrics, and MLLM-based assessments.

**World knowledge evaluation for text-to-image generation.** Evaluating text-to-image models with prompts that require world knowledge and commonsense reasoning has received increasing attention. WISE (Niu et al., 2025) evaluates complex text understanding and semantic reasoning grounded in cultural and scientific knowledge. RISEBench (Zhao et al., 2025) and KRIS-Bench (Wu et al., 2025) focus on assessing knowledge-based reasoning in image editing tasks. Several studies examine commonsense and physical reason-

ing: WorldGenBench (Zhang et al., 2025a) evaluates prompts involving implicit reasoning; R2I-Bench (Chen et al., 2025a) studies commonsense, mathematical, and logical reasoning; OmniGenBench (Yang et al., 2025) evaluates instruction-following capabilities involving physical and commonsense knowledge; and ABP (Zhang et al., 2025b) assesses implicit reasoning across knowledge of scientific, natural, and cultural scenes.

In contrast, KITTEN addresses a gap in concurrent benchmarks by emphasizing the reconstruction of detailed visual features (e.g., shape and color) for a broad set of real-world entities across diverse domains. We explicitly collect evaluation images for target entities and directly compare them with generated outputs, enabling a more precise assessment of visual-detail fidelity. Additionally, KITTEN evaluates retrieval-augmented models to determine whether explicitly providing reference images improves entity generation.

## 3    KITTEN Benchmark

We introduce the KITTEN benchmark to evaluate the reliability of text-to-image models in generating knowledge concepts.

### 3.1    Design Desiderata of KITTEN

The key to creating the benchmark is constructing a set of image-generation prompts that require grounding in visual-world knowledge. Two specific properties differentiate our benchmark from prior evaluation frameworks for image generation. First, while existing benchmarks aim to test the common-sense knowledge of image generation models, such as spatial or physical relationships (Gokhale et al., 2022; Huang et al., 2023), we focus on stress-testing the models by generating entities from specific domains. Therefore, we create the benchmark using image concepts from Wikipedia, a rich knowledge source that contains several domain-specific entities and their corresponding images. Second, while most existing benchmarks focus on evaluating the instruction-following capability of the models, we aim to understand how well these models faithfully represent real-world concepts grounded in visual knowledge sources. Therefore, we design a specific set of evaluations targeted at capturing the visual fidelity of generated entities. Guided by the above principles, we next clarify the details of the KITTEN benchmark.

### 3.2    Creating Entity-based Prompts

Fig. 2 shows the benchmark creation process. To generate a diverse set of prompts focused on faithfulness to knowledge-grounded concepts, we first select entity domains from the OVEN-Wiki dataset (Hu et al., 2023a), the most comprehensive open-domain image recognition dataset. We select eight domains comprising 322 entities, covering human-made objects, natural species, and human activities. This selection offers broader coverage than existing benchmarks (Lee et al., 2024). For each entity, we collect an evaluation set of entity images from Wikipedia for human evaluation, and a support set of images to evaluate retrieval-augmented models that leverage external knowledge sources for image generation (see evaluated models in Sec. 4).

After selecting the entities, we design five evaluation tasks for image-generation prompts.

- Basic prompt (4.58%): `Photo of Bandinelli Palace.`

- Entity in a specified location (30.57%): `Bandinelli Palace stands against snow mountains.`

- Composition with other objects (22.78%): `A peacock in front of Bandinelli Palace.`

- Entity in specific styles (21.20%): `An oil painting of Bandinelli Palace.`

- Entity made of specific materials (20.87%): `Bandinelli Palace made of crystal.`

For each task, ChatGPT (OpenAI et al., 2023) is instructed to generate text-to-image prompts for each entity domain according to the five evaluation tasks. These prompts are then tailored to evaluate knowledge-entity generation by incorporating entity names directly into the text. This process resulted in a set of 6,440 prompts to assess the models' ability to handle diverse and imaginative user inputs. The prompts range from 4 to 24 words, with an average length of 9.91 words and a standard deviation of 2.86.

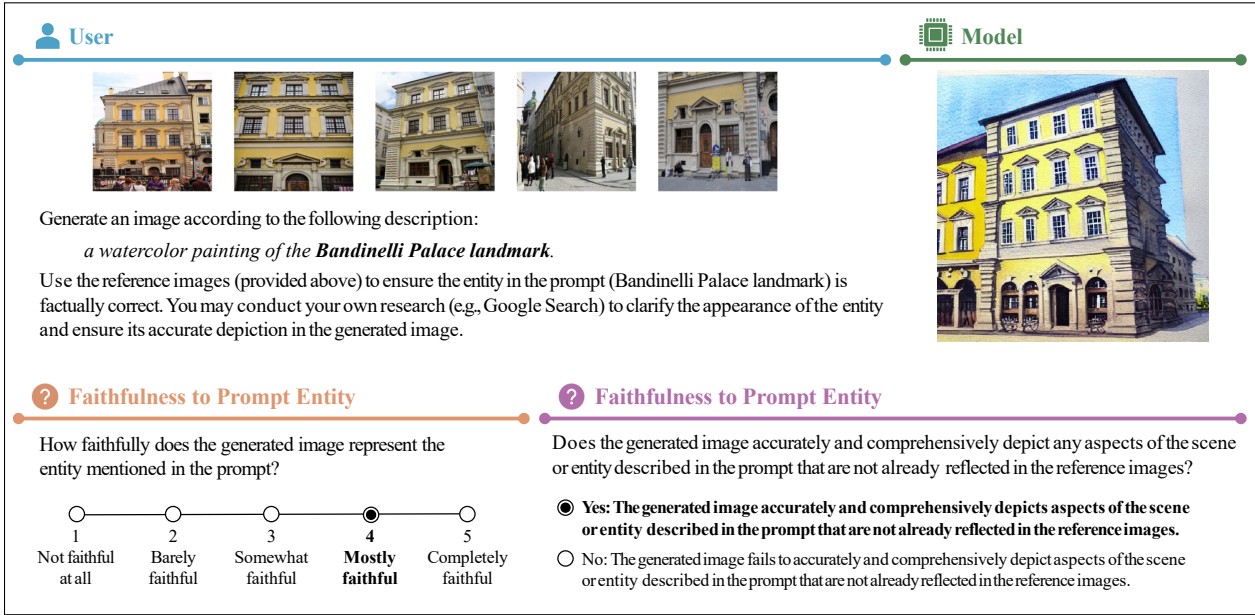

Figure 3: **Annotation interface.** Raters are asked to: 1) rate the image's faithfulness to the prompt entity on a 1–5 scale, and 2) indicate whether the image follows the prompt with a "yes" or "no" response. We report the percentage of responses marked as "yes."

## 3.3 Human Evaluation

Since no established metrics exist for evaluating the generation of visual entities, human evaluation plays a critical role in reliably assessing model performance. We design human evaluations to focus on the visual fidelity of the target entity in the generated image, decomposing the evaluation into two aspects: 1) faithfulness to the prompt entity, and 2) adherence to prompt instructions beyond the entity. This design allows raters to focus on distinct criteria, enabling more informative comparisons of models, as our results often show trade-offs between these aspects. Raters are shown reference images of the prompt entity and are encouraged to verify the entity's faithfulness through their own research. The final evaluation score is the average of five raters per image to ensure robust assessments. The human annotation interface is shown in Fig. 3.

- **Faithfulness to Entity.** We use a scale from 1 to 5, where 5 indicates complete faithfulness to the prompt entity, and 1 indicates that the generated image has no similarity to the prompt entity.

- **Instruction-following.** We use a yes/no question to evaluate whether the generated image adheres to the prompt instructions beyond the entity. The percentage of "yes" answers is then calculated.

## 3.4 Automatic Metrics

We gather the results of popular automatic metrics for image generation models, which primarily measure the similarity between the generated images and the evaluation images or prompts. While these metrics are not specialized for capturing the visual fidelity of the generated entity, we include their results to provide a comprehensive analysis of their alignment with human evaluation.

- **Image-Text Alignment.** We measure the cosine similarity between the generated image and the text prompt in CLIP's feature space (Radford et al., 2021), i.e., the *CLIP-T Score* (Hessel et al., 2021).

- **Image-Entity Alignment.** We measure the average pairwise cosine similarity between the generated image and the evaluation images of the target entity using DINO's feature space (Oquab et al., 2024). This serves as a proxy for how closely fine-grained details match.

### 3.5 MLLM Evaluation

Multimodal large language models (MLLMs) have recently demonstrated impressive progress in visual understanding. We investigate the use of MLLMs as automatic evaluators to reduce human effort and address the limitations of traditional metrics in assessing visual fidelity. Specifically, we prompt GPT-4o-mini (OpenAI et al., 2023) using the same criteria as our human evaluation.

- **MLLM Text Alignment.** Given the text prompt, we use an MLLM to evaluate whether the generated image follows the prompt instructions on a 1–5 scale.
- **MLLM Entity Alignment.** Given evaluation images of the target entity, we use an MLLM to assess how well the generated image resembles the target entity on a 1–5 scale.

## 4 Evaluated Models

We present a comprehensive analysis using KITTEN to understand the visual-world knowledge in current state-of-the-art models.

### 4.1 Text-to-Image Backbone Models

First, we examine general text-to-image backbone models that generate images solely based on text prompts, without using additional tools or reference images.

- Stable Diffusion (Rombach et al., 2022) maps images to a latent space where a diffusion model is trained. We mainly use SD-1.5 (Rombach et al., 2022) for a fair comparison with retrieval-augmented models, and also include SD-2.1 (Rombach et al., 2022), SD-3 (Esser et al., 2024), and SD-XL (Podell et al., 2023).
- Flux[1] (Black Forest Labs, 2024) is a successor to Stable Diffusion, integrating parallel transformer blocks.
- Imagen (Saharia et al., 2022) uses a T5 encoder and cascaded diffusion models for high-resolution image generation.
- Imagen-3 (Imagen 3 Team, 2024) is a successor to Imagen, notable for its ability to handle long prompts.
- DALL·E-2 (Ramesh et al., 2022) is a diffusion model conditioned on CLIP embeddings, notable for its zero-shot compositional abilities.

### 4.2 Retrieval-augmented Text-to-Image Models

Retrieval-augmented methods are a family of image generation approaches that use support images (e.g., retrieved by a search engine) to enhance the model through fine-tuning or in-context learning, improving the fidelity of entities in generated images. Our goal is to evaluate these models and determine whether incorporating such *support* images enhances the fine-grained visual fidelity of the entity during generation. Specifically, we provide ground-truth reference entity images (held out from the entity images used for evaluation) as the support images to these methods and then generate new images from them following the evaluation text prompts. We study the following models:

- DreamBooth (Ruiz et al., 2023) **fine-tunes** the SD-1.5 model to learn a special token encoding the target entity. It then generates the entity in new contexts using prompts that include this token.
- Custom-Diff (Kumari et al., 2023), similar to DreamBooth, **fine-tunes** partial weights of SD-1.5.
- Instruct-Imagen (Hu et al., 2024) generates the target entity through **in-context learning** by encoding support images into a multimodal instruction: `Generate an image of <entity_name>, referring to the images <ref_image_1>, ..., <ref_image_K>, and follow the caption: <prompt>.`
- BLIP-Diffusion (Li et al., 2023) performs **in-context learning** by training a multimodal encoder to obtain support image embeddings aligned with text prompt embeddings.
- IP-Adapter (Ye et al., 2023) performs **in-context learning** by training separate cross-attention layers to incorporate support images as inputs.

---

[1]We use Flux.1-dev.

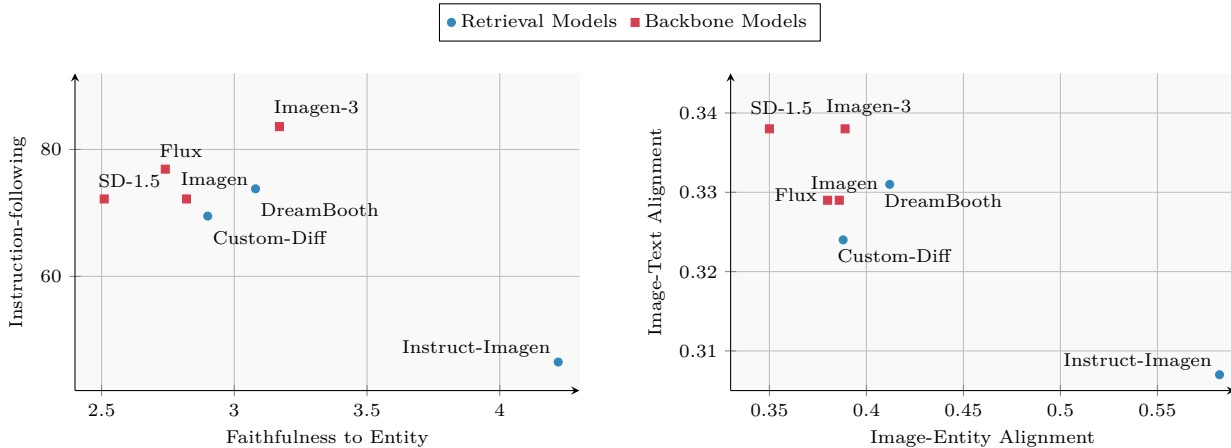

Figure 4: **Evaluation results of text-to-image models. (Left)** Human evaluation highlights the trade-off between entity faithfulness and instruction-following. **(Right)** Automatic metrics illustrate the relationship between image-entity and image-text alignment.

### 4.3 Unified Understanding and Generation Models

Finally, we evaluate unified models that jointly learn multimodal understanding (e.g., image captioning, visual question answering) and generation tasks (e.g., text-to-image generation) within the same framework. These models do not take support images as inputs. We study the following models:

- Show-o (Xie et al., 2024) tokenizes multimodal data into a sequence, processing text tokens autoregressively and image tokens via discrete diffusion modeling.
- Janus-Pro (Chen et al., 2025b) employs a unified transformer with autoregressive prediction, while using separate visual encoders for the understanding and generation components.
- Emu3 (Wang et al., 2024) tokenizes multimodal data into discrete token spaces and trains a single transformer on next-token prediction tasks.

## 5 Evaluation Results

### 5.1 Human Evaluation

**Retrieval-augmented models enhance faithfulness but weaken instruction-following.** Fig. 4 (left) shows that retrieval-augmented models, including Custom-Diff, DreamBooth, and Instruct-Imagen, generally produce images that are more faithful to the entities than their base models, SD-1.5 and Imagen. This is because these models incorporate reference images during testing, enabling them to generate visual concepts that are not well represented in the base models' parameters. However, retrieval-augmented models tend to have reduced instruction-following capabilities, as they often over-rely on reference images and struggle to create novel configurations of the entity, as requested in creative text prompts. Although this trend is consistent across methods, the extent of the impact varies. Notably, Instruct-Imagen shows a significant increase in faithfulness (2.81 → 4.22) but also a substantial drop in instruction-following (72.2 → 46.5).

**Enhancing backbone models improves both faithfulness and instruction-following.** For example, Flux outperforms its predecessor SD-1.5 by 0.23, and Imagen-3 surpasses its predecessor Imagen by 0.35 in faithfulness. However, these improvements in faithfulness are still minor compared to those achieved by retrieval-augmented methods, such as DreamBooth, which shows a 0.57 improvement over SD-1.5.

On the other hand, Imagen-3 achieves the highest instruction-following score (83.6). A notable strength is its strong compositional capability in generating challenging scenarios, such as an insect wearing sunglasses. Imagen-3 also attains a high faithfulness score (3.17), outperforming the retrieval-augmented models DreamBooth (3.08) and Custom-Diff (2.90). This demonstrates that Imagen-3, as a strong backbone model,

Table 1: **Evaluation results.** MLLM evaluation shows the relationship between entity and text alignment.

| Metrics | Backbone Models | | | | | | | |
|---|---|---|---|---|---|---|---|---|
| | DALL·E-2 | SD-1.5 | SD-2.1 | SD-3 | SD-XL | Imagen | Flux | Imagen-3 |
| MLLM Entity Alignment | 2.65 | 2.39 | 2.45 | 2.77 | 2.79 | 2.53 | 2.04 | 2.83 |
| MLLM Text Alignment | 3.69 | 3.46 | 3.66 | 4.07 | 4.19 | 3.61 | 3.96 | 4.17 |
| Metrics | Retrieval Models | | | | | Unified Models | | |
| | BLIP-Diffusion | IP-Adapter | Custom-Diff | DreamBooth | Instruct-Imagen | Show-o | Janus-Pro | Emu3 |
| MLLM Entity Alignment | 2.52 | 2.82 | 2.70 | 2.82 | 3.72 | 1.93 | 1.89 | 2.22 |
| MLLM Text Alignment | 2.08 | 2.65 | 3.29 | 3.37 | 2.63 | 3.55 | 3.65 | 3.96 |

can generate specialized entities solely from text prompts. However, Imagen-3 still faces challenges with low-frequency entities, and a notable gap remains in entity fidelity compared to the highest score achieved by Instruct-Imagen (4.22). These findings show that enhancing the backbone model can improve both instruction-following and entity fidelity, while it is essential to incorporate advanced retrieval-augmented techniques to achieve even higher levels of faithfulness.

**Balancing faithfulness and instruction-following is achievable.** The retrieval-augmented model DreamBooth improves entity faithfulness compared to its baseline, SD-1.5 (2.51 → 3.08), without compromising SD-1.5's instruction-following score (72.2 → 73.8). This demonstrates that a well-designed retrieval-augmented method can enhance entity fidelity without sacrificing creativity. These findings also suggest future research directions, emphasizing that combining a strong backbone with an effective retrieval-augmented approach can achieve a balance between faithfulness and instruction-following.

## 5.2 Automatic Metrics

**Retrieval-augmented models improve entity alignment but reduce text alignment.** This observation, shown in Fig. 4 (right), is consistent with the human evaluation results. Moreover, this trend aligns with findings in recent work (Materzyńska et al., 2023), which indicate that models incorporating additional inputs, such as reference images, tend to have lower text alignment scores than base models. This reflects a trade-off between aligning with the text and aligning with the images.

**Alignment between automatic metrics and human evaluation.** While the overall trends from the automatic metrics align with the human evaluation, there are notable discrepancies. We observe that improving base models does not necessarily lead to gains in the automatic metrics. For example, Flux (0.329) performs worse than its predecessor SD-1.5 (0.338) in the image-text metric, and Imagen-3 (0.389) shows only a marginal improvement over Imagen (0.386) in the image-entity score. These findings suggest that automatic metrics have a limited ability to capture meaningful variations between models of similar quality. In addition, although DreamBooth achieves a higher instruction-following score compared to its base model SD-1.5, it has a lower image-text alignment score. We hypothesize that the image-text score may not accurately assess the alignment between the generated image and rare entities. For example, with the prompt "The Teufelsmauer landmark shimmers in the sunlight," it is unclear whether the image-text similarity for "Teufelsmauer" is evaluated correctly. This highlights that traditional metrics (Hessel et al., 2021; Lee et al., 2024) might fail to measure true alignment between the unique entity and the generated image. Furthermore, Imagen-3 ranks higher in faithfulness in the human evaluation, yet DreamBooth outperforms Imagen-3 in image-entity scores, indicating a misalignment between human perception and the learned semantic features (Oquab et al., 2024).

## 5.3 MLLM Evaluation

**Alignment between MLLM and human evaluation.** Tab. 1 shows the results of the MLLM evaluation. The MLLM scores generally exhibit a high correlation with human evaluation results shown in Fig. 4 (left), although slight discrepancies exist. Specifically, DreamBooth (3.37) falls behind SD-1.5 (3.46) and Imagen (3.61) in the MLLM text alignment score. However, in human evaluation, DreamBooth (73.8) outperforms both Imagen and SD-1.5 (72.2). This suggests that while the MLLM can capture overall trends and identify models with clearly stronger or weaker performance, it may struggle to distinguish between models with similar capabilities. On the other hand, Flux (2.04) receives a lower MLLM entity alignment score than

SD-1.5 (2.39), despite outperforming SD-1.5 in human evaluation (2.74 vs. 2.51). This indicates that the MLLM may have different preferences or biases compared to human raters.

**Gradual improvements in the SD series.** SD-1.5, SD-2.1, SD-3, and SD-XL show consistent improvements in both MLLM entity and text alignment scores. Notably, SD-XL achieves strong performance comparable to Imagen-3, with similar entity (2.79 vs. 2.83) and text alignment scores (4.19 vs. 4.17). These models generate high-frequency entities accurately but struggle with low-frequency entities, where the generated examples often fail to match the target's attributes or deviate significantly in structure and configuration.

**In-context methods show weaker text alignment than fine-tuning methods.** For retrieval-based models that rely on in-context learning, BLIP-Diffusion underperforms Imagen in entity alignment (2.53) and lags substantially in text alignment (2.08 vs. 3.61). IP-Adapter achieves a higher entity alignment score, close to DreamBooth (2.82), but exhibits a lower text alignment score (2.65 vs. 3.37). These results indicate that in-context learning approaches generally have weaker instruction-following capabilities than fine-tuning-based methods, such as DreamBooth and Custom-Diff. We observe that in-context learning models often almost directly replicate the reference images in their outputs. While they maintain high entity fidelity, they struggle to follow instructions, rarely generate new materials or styles, and often fail to incorporate additional objects.

**Unified models struggle to generate accurate visual entities.** The unified models Show-o (3.55) and Janus-Pro (3.65) achieve moderate text alignment scores, comparable to Imagen (3.61). However, all three unified models exhibit noticeably lower entity alignment scores, even below SD-1.5 (2.39). For low-frequency categories, such as plants and insects, these models often generate entities that bear little resemblance to the target. While high-frequency entities are produced more reliably, the models still struggle with common items, such as Peking duck and macarons, highlighting persistent challenges in accurate entity generation. These results suggest that, although unified models may gain broader world knowledge through joint training on understanding and generation tasks, effectively translating this knowledge into accurate visual entity generation remains a significant challenge.

### 5.4 Ablation Study

**Selection of image-entity alignment metrics.** In Tab. 2, two popular visual features are tested for calculating cosine similarity scores between reference and generated images as the image-entity alignment metric: *CLIP-I* (Radford et al., 2021) and *DINO* (Oquab et al., 2024). DINO scores provide a clearer separation between models compared to CLIP-I scores, making them a more discriminative metric for capturing subtle differences in faithfulness. For example, the difference between Custom-Diff and Instruct-Imagen is much larger when using DINO (0.19) compared to CLIP-I (0.11). This may be due to DINO's focus on primary entities, which allows for a more accurate estimation of similarity between the generated entities and reference images.

Table 2: Selection of image-entity metrics.

| Models | CLIP-I | DINO |
|---|---|---|
| SD-1.5 | 0.646 | 0.350 |
| Imagen | 0.646 | 0.386 |
| Flux | 0.639 | 0.380 |
| Imagen-3 | 0.650 | 0.389 |
| Custom-Diff | 0.643 | 0.388 |
| DreamBooth | 0.674 | 0.412 |
| Instruct-Imagen | 0.751 | 0.582 |

**Correlation of automatic metrics with human evaluation.** While manual evaluation ensures high accuracy, automated methods provide a cost-effective alternative, albeit with slightly lower alignment to human perception. To quantify the consistency between automatic metrics and human evaluation, we computed Pearson and Spearman correlations in Tab. 3. CLIP-T and CLIP-I show moderate alignment with user evaluations, while DINO demonstrates stronger alignment with human judgments of faithfulness compared to CLIP-I.

Table 3: Correlation of automatic and MLLM metrics with human evaluation.

| Metrics | Models | Pearson | Spearman |
|---|---|---|---|
| Entity Alignment | GPT-4o | **0.703** | **0.695** |
| | Qwen-2.5 | 0.531 | 0.520 |
| | DINO | 0.510 | 0.504 |
| | CLIP-I | 0.239 | 0.340 |
| Text Alignment | GPT-4o | **0.618** | **0.589** |
| | Qwen-2.5 | 0.515 | 0.513 |
| | CLIP-T | 0.337 | 0.384 |

**Correlation of MLLM metrics with human evaluation.**
We assess the correlation between MLLM-based evaluation and human judgments using two different MLLMs, GPT-4o (OpenAI et al., 2023) and Qwen-2.5 (Team, 2025). MLLM-based evaluation shows strong alignment with human preferences, highlighting its potential to capture subtle visual details when assessing the faithfulness of generated knowledge entities. This demonstrates that our evaluation framework can be reliably scaled using MLLMs as automatic evaluators. Furthermore, GPT-4o achieves significantly higher correlations

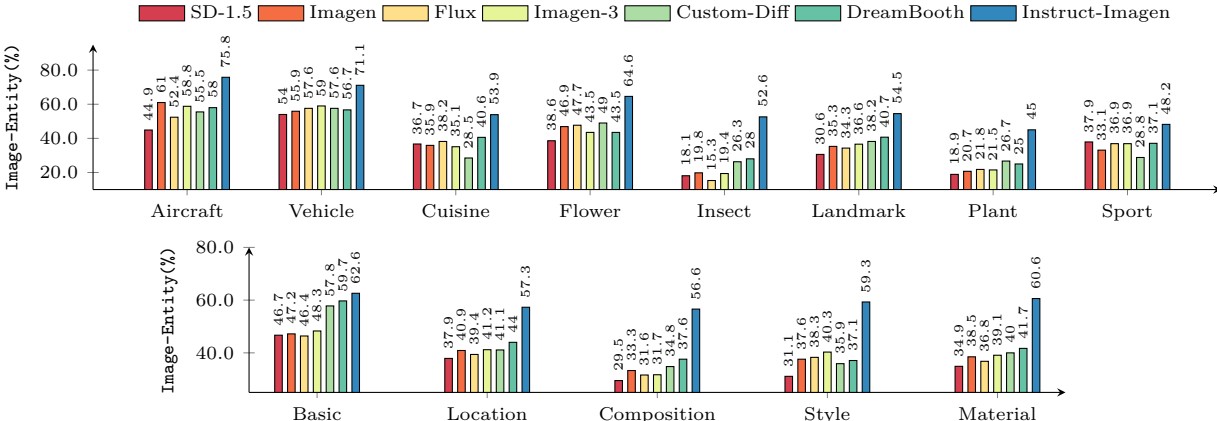

Figure 5: **Performance analysis across domains and tasks. (Top)** Retrieval models achieve higher image-entity scores in the insect, landmark, and plant domains but perform worse in the cuisine and sport domains, possibly due to the varying occurrence of these entities in common image datasets. **(Bottom)** *Location* scores the highest, while *Composition* scores the lowest.

with human evaluation compared to Qwen-2.5, while Qwen-2.5 still outperforms traditional CLIP and DINO metrics. These results indicate that our benchmark can be effectively extended using open-source MLLMs for automatic evaluation.

## 5.5 Analysis of Performance Variations

**Performance across entity domains.** Fig. 5 (top) shows that the performance of each method is domain-dependent. To analyze how domain distinctions affect model performance, we categorize the eight domains in our dataset along a spectrum of visual variability and frequency in the training data: 1) *Generic classes* (e.g., sports, cuisine) exhibit the highest inter-category variation, with items differing substantially in appearance, and appear most frequently in the training data. 2) *Specific classes* (e.g., insects, flowers) show moderate variation, as items within the same species can still look distinct, and occur with moderate frequency. 3) *Generic instances* (e.g., aircraft, vehicles) have lower variation, as different instances of the same model share similar structures, and also occur moderately often. 4) *Specific instances* (e.g., landmarks) exhibit the lowest variation, with a single, fixed visual appearance, and appear least frequently.

For specific instances, retrieved images are highly representative, enabling retrieval-augmented models to produce outputs that closely align with the targets and achieve higher scores than backbone models. Since these concepts are less frequent in common image datasets, they are underrepresented in the backbone models' parameters. Thus, incorporating reference images during inference improves the performance of retrieval-augmented models. In contrast, for generic classes, retrieved images exhibit higher variation and are less representative, so outputs from retrieval-augmented models often resemble the target images less closely, resulting in lower performance compared to backbone models. In these domains, where concepts are common and well-memorized by the base models, retrieval may even harm performance due to overfitting on limited reference images. Overall, this variability suggests that the effectiveness of retrieval-augmented methods depends on the nature of the domain-specific content, and that the optimal choice of retrieval strategy remains an open question.

**Performance across evaluation tasks.** Fig. 5 (bottom) shows that image-entity scores across evaluation tasks generally align with the overall ranking. *Location* scores highest (0.431), followed by *Material* (0.417), *Style* (0.399), and *Composition* (0.364), highlighting the challenge of maintaining entity fidelity when prompts involve complex compositions.

We observe that *Style* prompts show a distinct score distribution. Retrieval-augmented methods, DreamBooth and Custom-Diff, along with their base model SD-1.5, receive lower image-entity scores (0.371, 0.359, and 0.311), indicating that models based on SD-1.5 struggle to generate faithful entities when changing their

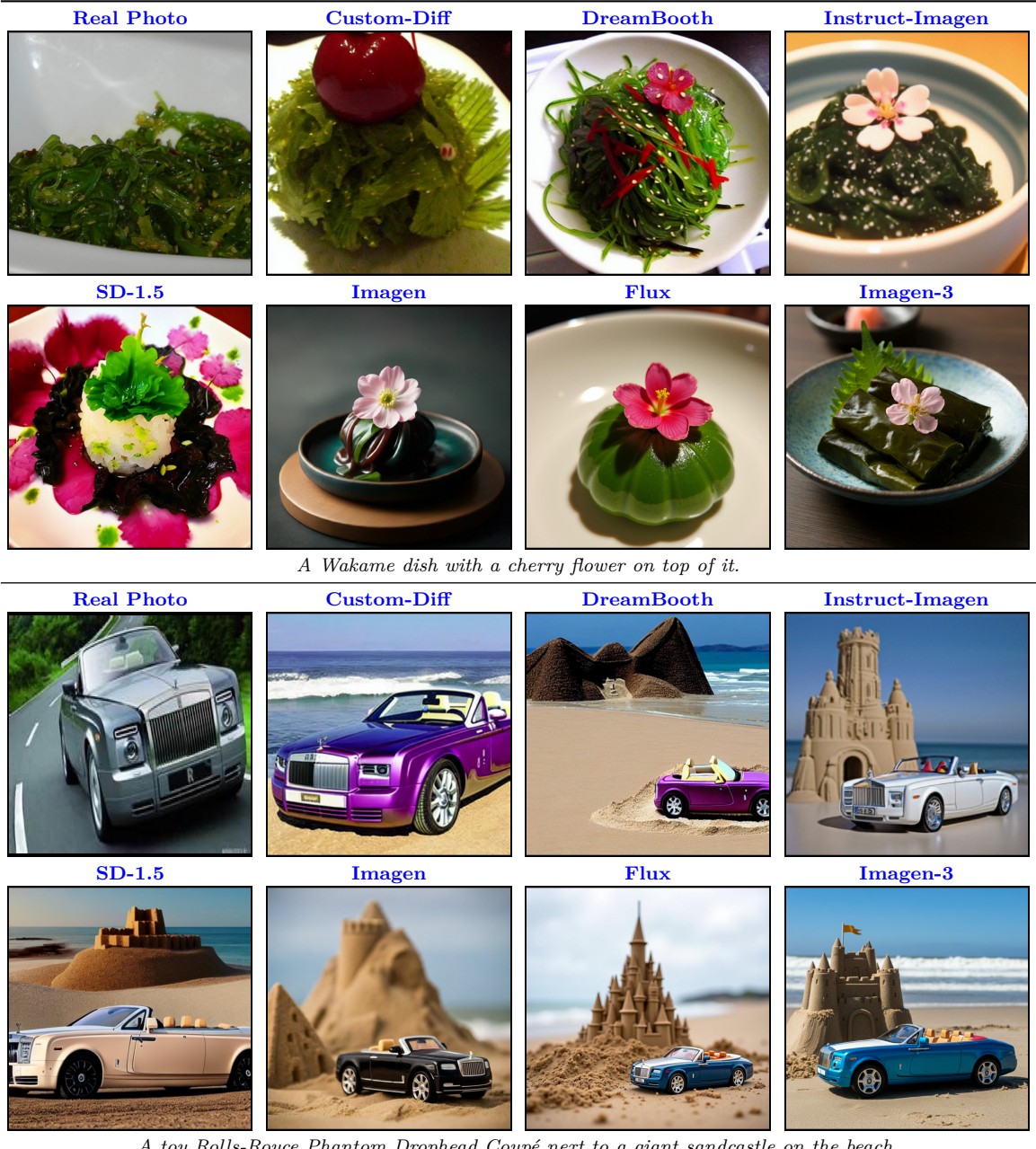

Figure 6: **Qualitative results. (Top)** The backbone models, including SD-1.5, Imagen, Flux, and Imagen-3, show lower faithfulness to the entity, with Wakame's appearance differing from the reference image. **(Bottom)** The retrieval models, including Custom-Diff, DreamBooth, and Instruct-Imagen, struggle with instruction following and fail to create a composition between the entity and the giant sandcastle.

styles. However, these models achieves the highest image-text score (0.346, 0.341, and 0.335), suggesting they are strong in generating accurate styles but may sacrifice entity fidelity.

### 5.6 Qualitative Results

We present the visual results in Fig. 6. In the above example, the backbone models (second row) show lower faithfulness to the entity, with Wakame's appearance differing from the reference image. In contrast,

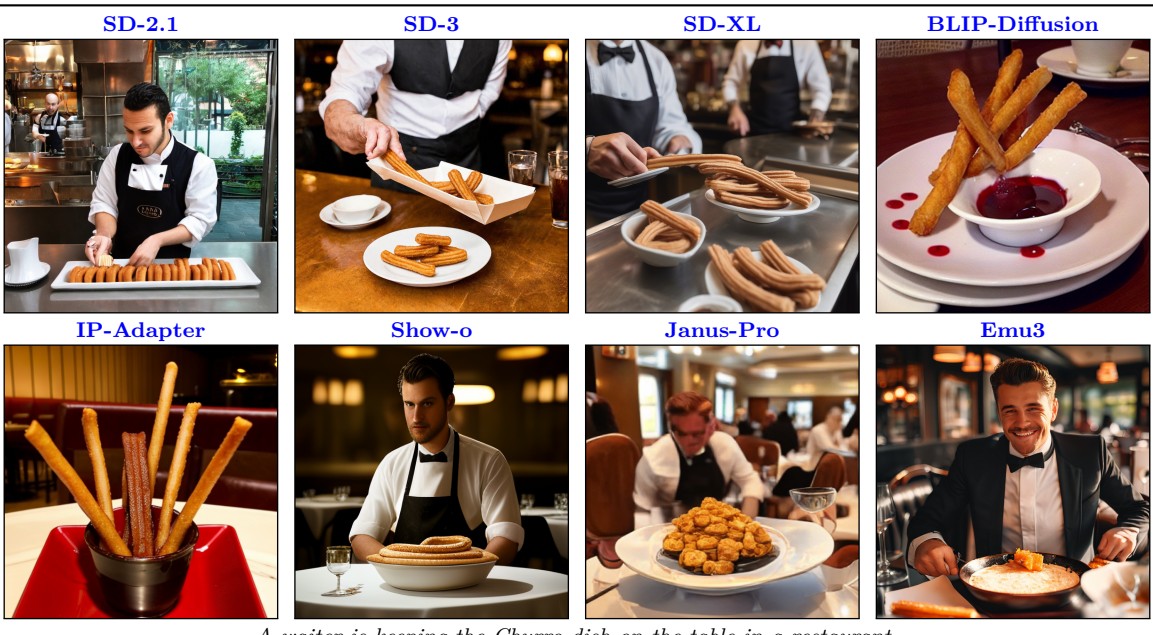

| SD-2.1 | SD-3 | SD-XL | BLIP-Diffusion |
| IP-Adapter | Show-o | Janus-Pro | Emu3 |

*A waiter is keeping the Churro dish on the table in a restaurant.*

Figure 7: **Additional qualitative results.** The SD series, including SD-2.1, SD-3, and SD-XL, shows gradual improvements in faithfulness to the entity. Retrieval-augmented models based on in-context learning, such as BLIP-Diffusion and IP-Adapter, achieve strong entity fidelity but fail to follow the prompt. Unified models, including Show-o, Janus-Pro, and Emu3, generate incorrect visual details of the target entity.

retrieval-augmented models (first row), which use reference images during testing, achieve better visual alignment with the target. Instruct-Imagen demonstrates a balance between entity fidelity and creative flexibility in the generated images. In the example below, Custom-Diff and DreamBooth exhibit reduced instruction-following, struggling with compositional prompts and often omitting the main entity or secondary objects. In contrast, the backbone models excel in entity faithfulness, likely because the target entity is well-represented in their training data. Retrieval-augmented models underperform when they over-rely on reference images and experience knowledge forgetting during fine-tuning on small reference sets.

We show additional results in Fig. 7. The SD series demonstrates gradual improvements in capturing the correct details of the target entity. Retrieval-augmented models based on in-context learning (e.g., BLIP-Diffusion and IP-Adapter) achieve strong entity fidelity but struggle to follow the prompt, failing to generate the waiter. Unified models (e.g., Show-o, Janus-Pro, Emu3) produce incorrect visual details of the target entity, Churro.

## 6 Conclusion

We propose KITTEN, a benchmark for evaluating entity fidelity in text-to-image generation, focusing on visual concepts that require specialized knowledge. We design prompts based on Wikipedia entities and introduce a human and MLLM evaluation framework to assess visual faithfulness. Extensive analysis reveals that while backbone models can generate specialized entities, retrieval-augmented models achieve higher fidelity. However, these methods often struggle with creative prompts, highlighting the need for techniques that enhance entity fidelity without compromising instruction-following ability.

**Limitations and Future Work.** We leave the evaluation of prompts requiring implicit reasoning (e.g., "the tallest building in Manhattan") for future work, as such prompts can often be reformulated into straightforward, entity-focused prompts (e.g., "One World Trade Center") through rewriting.

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
