# 🐱 KITTEN: A Knowledge-Integrated Evaluation of Image Generation on Visual Entities

**Hsin-Ping Huang**[1,2*]   **Xinyi Wang**[1*]   **Yonatan Bitton**[1]   **Hagai Taitelbaum**[1]
**Gaurav Singh Tomar**[1]   **Ming-Wei Chang**[1]   **Xuhui Jia**[1]   **Kelvin C.K. Chan**[1]   **Hexiang Hu**[1]
**Yu-Chuan Su**[1]   **Ming-Hsuan Yang**[1,2]

[1]**Google DeepMind**     [2]**University of California, Merced**

Reviewed on OpenReview: **https://openreview.net/forum?id=wejaKS9Ps0**

## 1   Additional Qualitative Examples

We present additional qualitative examples evaluating the backbone models, SD-1.5, Imagen, Flux, and Imagen-3, as well as the retrieval models, Custom-Diff, DreamBooth, and Instruct-Imagen, across various domains in the KITTEN benchmark. The results for the aircraft, vehicle, cuisine, flower, insect, landmark, plant, and sport domains are shown in Fig. 1, Fig. 2, Fig. 3, Fig. 4, Fig. 5, Fig. 6, Fig. 7, Fig. 8, respectively.

Fig. 3 presents the results for the cuisine domain. For the prompt, "A Wakame dish with a cherry flower on top of it," retrieval-based models such as Custom-Diff, DreamBooth, and Instruct-Imagen successfully generate the entity Wakame. However, Custom-Diff fails to capture the composition involving the cherry flower. In contrast, the backbone models, Imagen, Flux, and Imagen-3, correctly generate the cherry flower, but the representation of Wakame does not align with its real-world appearance. In the example prompt, "photo of a hot and sour soup," the backbone models introduce incorrect ingredients that significantly deviate from a real-world hot and sour soup, demonstrating how these models hallucinate entities instead of reproducing real-world knowledge accurately.

Fig. 5 shows the results for the insect domain. For the prompt, "Satyrium liparops sitting at the beach with a view of the sea," all models correctly generate the beach scene as instructed. However, the backbone models hallucinate the insect's appearance, generating a completely different insect, while the retrieval models accurately depict the visual details of *Satyrium liparops.* For the prompt, "Promachus hinei wearing sunglasses," the backbone models demonstrate strong instruction-following ability, with Imagen and Imagen-3 correctly composing the sunglasses on the insect. In contrast, the retrieval models fail to generate this composition. However, the backbone models do not generate an accurate representation of the insect itself, while the retrieval models correctly generate the entity.

Fig. 7 presents the results for the plant domain. In the example, "An impressionistic painting of Cirsium andersonii," none of the models effectively follow the prompt in generating the "impressionistic painting," highlighting the challenge of rendering specific artistic styles. However, the retrieval models generate the entity more accurately than the backbone models. In another example, "Penstemon rydbergii on a rustic wooden table next to a rose plant," although the generated entity lacks the fine details of the reference, the backbone models successfully capture the composition of "next to a rose," demonstrating their stronger instruction-following ability.

These observations align with the quantitative results, further demonstrating that while advanced backbone models can generate specialized entities, there is a notable gap in entity fidelity compared to retrieval-augmented models. Although retrieval-augmented methods improve faithfulness to entities, they often struggle with creative prompts, highlighting the need for techniques that enhance entity accuracy without compromising instruction-following abilities.

---

*Equal contribution

In addition, we have provided examples illustrating a balance between entity fidelity and creative flexibility in the generated images. Specifically, Instruct-Imagen's results for prompts such as "A Wakame dish with a cherry flower on top of it" in Fig. 3, "Alstroemeria on top of a mountain with sunrise in the background" in Fig. 4, and "Satyrium liparops sitting at the beach with a view of the sea" in Fig. 5 demonstrate enhanced entity fidelity while maintaining creative flexibility. These examples are supported by the highest alignment scores across both metrics when compared to other models.

## 2 Details of the Human Evaluation Scores Across Domains

We have presented the human evaluation results for backbone and retrieval-augmented text-to-image models, comparing their performance in terms of entity faithfulness and instruction-following accuracy. Here, we provide a detailed breakdown of the human evaluation results across eight domains of the KITTEN benchmark in Tab. 1.

We observe that the performance of each method varies across domains. For instance, while the overall faithfulness scores of Custom-Diff and DreamBooth are lower than those of the backbone model Imagen-3, these retrieval models consistently perform better in the insect, landmark, and plant domains. When comparing backbone models, Flux outperforms SD-1.5 in the overall score; however, SD-1.5 demonstrates higher faithfulness in the insect, landmark, and plant domains, suggesting that Flux struggles to generate less frequent entities. Additionally, the retrieval models, DreamBooth and Custom-Diff, improve faithfulness over SD-1.5 in some cases but show reduced performance in the cuisine and sport domains. In terms of instruction-following scores, Instruct-Imagen performs notably lower in the flower (31.3) and insect (21.9) domains, while Imagen-3 achieves significantly higher scores in the cuisine (93.8) and sport (96.9) domains.

Table 1: Detailed breakdown of human evaluation.

| Model | Aircraft | Vehicle | Cuisine | Flower | Insect | Landmark | Plant | Sport | Overall |
|---|---|---|---|---|---|---|---|---|---|
| Faithfulness to entity | | | | | | | | | |
| ■ SD | 2.07 | 3.57 | 3.42 | 2.36 | 1.61 | 1.95 | 2.09 | 3.02 | 2.51 |
| ■ Imagen | 3.19 | 3.62 | 3.55 | 3.21 | 1.54 | 2.11 | 2.04 | 3.3 | 2.82 |
| ■ Flux | 3.51 | 3.89 | 3.35 | 2.52 | 1.44 | 1.77 | 1.83 | 3.59 | 2.74 |
| ■ Imagen-3 | 3.74 | 4.04 | 4.26 | 3.37 | 1.63 | 2.31 | 1.98 | 4.04 | 3.17 |
| ● Custom-Diff | 2.59 | 3.90 | 2.52 | 3.82 | 2.37 | 3.02 | 2.85 | 2.10 | 2.90 |
| ● DreamBooth | 3.26 | 3.88 | 3.23 | 3.18 | 2.46 | 3.06 | 2.64 | 2.93 | 3.08 |
| ● Instruct-Imagen | 4.23 | 4.70 | 4.43 | 4.48 | 3.78 | 4.06 | 4.02 | 4.08 | **4.22** |
| Instruction-following | | | | | | | | | |
| ■ SD | 68.8 | 75.0 | 62.5 | 87.5 | 62.5 | 65.6 | 84.4 | 71.9 | 72.2 |
| ■ Imagen | 59.4 | 84.4 | 75.0 | 84.4 | 53.1 | 59.4 | 84.4 | 78.1 | 72.2 |
| ■ Flux | 56.3 | 87.5 | 96.9 | 78.1 | 59.4 | 71.9 | 84.4 | 81.3 | 76.9 |
| ■ Imagen-3 | 78.1 | 87.5 | 93.8 | 84.4 | 71.9 | 71.9 | 84.4 | 96.9 | 83.6 |
| ● Custom-Diff | 59.4 | 75.0 | 65.6 | 78.1 | 71.9 | 65.6 | 81.3 | 59.4 | 69.5 |
| ● DreamBooth | 68.8 | 71.9 | 71.9 | 81.3 | 68.8 | 75.0 | 78.1 | 75.0 | 73.8 |
| ● Instruct-Imagen | 46.9 | 56.3 | 56.3 | 31.3 | 21.9 | 50.0 | 59.4 | 50.0 | 46.5 |

(■: Text-to-Image Models, ●: Retrieval-augmented Models)

## 3 Details of the Automatic Metric Scores Across Domains

We have presented the automatic metric evaluation for backbone and retrieval-augmented text-to-image models, including image-text alignment and image-entity alignment scores. Here, we include a detailed breakdown of the automatic metrics across eight domains of the KITTEN benchmark in Tab. 2. Specifically, we include a detailed ablation study of the image-entity alignment metric using cosine similarity scores based on two popular image features: CLIP-Image and DINO features.

The retrieval models, Custom-Diff, DreamBooth, and Instruct-Imagen, consistently outperform the backbone models in DINO scores across the insect, landmark, and plant domains. In contrast, Custom-Diff shows notably worse DINO and CLIP-T scores in the cuisine and sport domains. These results are consistent with the human evaluations.

Table 2: Detailed breakdown of automatic metrics.

| Model | Aircraft | Vehicle | Cuisine | Flower | Insect | Landmark | Plant | Sport | Overall |
|---|---|---|---|---|---|---|---|---|---|
| Image-Text Alignment: CLIP-T | | | | | | | | | |
| ■ SD | 0.341 | 0.354 | 0.343 | 0.336 | 0.322 | 0.330 | 0.332 | 0.342 | 0.338 |
| ■ Imagen | 0.329 | 0.344 | 0.338 | 0.337 | 0.323 | 0.312 | 0.316 | 0.335 | 0.329 |
| ■ Flux | 0.341 | 0.346 | 0.329 | 0.327 | 0.320 | 0.308 | 0.314 | 0.343 | 0.329 |
| ■ Imagen-3 | 0.345 | 0.351 | 0.352 | 0.343 | 0.325 | 0.315 | 0.324 | 0.352 | 0.338 |
| ● Custom-Diff | 0.333 | 0.349 | 0.314 | 0.339 | 0.321 | 0.315 | 0.326 | 0.298 | 0.324 |
| ● DreamBooth | 0.333 | 0.347 | 0.335 | 0.333 | 0.322 | 0.318 | 0.324 | 0.335 | 0.331 |
| ● Instruct-Imagen | 0.293 | 0.324 | 0.315 | 0.320 | 0.295 | 0.294 | 0.302 | 0.316 | 0.307 |
| Image-Entity Alignment: CLIP-I | | | | | | | | | |
| ■ SD | 0.640 | 0.655 | 0.673 | 0.719 | 0.619 | 0.616 | 0.628 | 0.618 | 0.646 |
| ■ Imagen | 0.678 | 0.650 | 0.671 | 0.733 | 0.630 | 0.632 | 0.630 | 0.545 | 0.646 |
| ■ Flux | 0.652 | 0.653 | 0.663 | 0.738 | 0.609 | 0.614 | 0.618 | 0.561 | 0.639 |
| ■ Imagen-3 | 0.677 | 0.656 | 0.665 | 0.726 | 0.623 | 0.638 | 0.632 | 0.582 | 0.650 |
| ● Custom-Diff | 0.688 | 0.678 | 0.593 | 0.759 | 0.619 | 0.655 | 0.662 | 0.491 | 0.643 |
| ● DreamBooth | 0.698 | 0.676 | 0.692 | 0.742 | 0.660 | 0.681 | 0.657 | 0.589 | 0.674 |
| ● Instruct-Imagen | 0.776 | 0.719 | 0.746 | 0.825 | 0.810 | 0.735 | 0.751 | 0.648 | 0.751 |
| Image-Entity Alignment: DINO | | | | | | | | | |
| ■ SD | 0.449 | 0.540 | 0.367 | 0.386 | 0.181 | 0.306 | 0.189 | 0.379 | 0.350 |
| ■ Imagen | 0.610 | 0.559 | 0.359 | 0.469 | 0.198 | 0.353 | 0.207 | 0.331 | 0.386 |
| ■ Flux | 0.524 | 0.576 | 0.382 | 0.477 | 0.153 | 0.343 | 0.218 | 0.369 | 0.380 |
| ■ Imagen-3 | 0.588 | 0.590 | 0.351 | 0.435 | 0.194 | 0.366 | 0.215 | 0.369 | 0.389 |
| ● Custom-Diff | 0.555 | 0.576 | 0.285 | 0.490 | 0.263 | 0.382 | 0.267 | 0.288 | 0.388 |
| ● DreamBooth | 0.580 | 0.567 | 0.406 | 0.435 | 0.280 | 0.407 | 0.250 | 0.371 | 0.412 |
| ● Instruct-Imagen | 0.758 | 0.711 | 0.539 | 0.646 | 0.526 | 0.545 | 0.450 | 0.482 | 0.582 |

(■: Text-to-Image Models, ●: Retrieval-augmented Models)

## 4 Details of the MLLM Evaluation Scores Across Domains

We report MLLM evaluation results for both backbone and retrieval-augmented text-to-image models, including scores for text alignment and entity alignment. Tab. 3 presents a detailed breakdown of these scores across the eight domains of the KITTEN benchmark. Notably, retrieval-augmented models consistently outperform backbone models in entity alignment within the vehicle, insect, landmark, and plant domains. In contrast, Custom-Diff shows substantially lower entity alignment scores in the cuisine and sport domains.

## 5 Details of the Human Evaluation Scores Across Prompts

We provide a detailed breakdown of human evaluation results across five evaluation tasks in the KITTEN benchmark, as shown in Tab. 4. For the faithfulness score, the ranking across prompts remains consistent with the overall ranking. Instruct-Imagen achieves the highest score, followed by Imagen-3. The retrieval-based methods, DreamBooth and Custom-Diffusion, come next, while the other base models, Imagen, Flux, and Stable-Diffusion, score comparatively lower.

The instruction-following scores across prompts align with the average performance, with Imagen-3 emerging as the top performer. Notably, Flux and Imagen-3 excel in *Location*, achieving scores of 90 and 87.8, respectively,

Table 3: Detailed breakdown of MLLM evaluation.

| Model | Aircraft | Vehicle | Cuisine | Flower | Insect | Landmark | Plant | Sport | Overall |
|---|---|---|---|---|---|---|---|---|---|
| MLLM Text Alignment | | | | | | | | | |
| ■ SD | 3.06 | 3.62 | 2.97 | 4.22 | 2.84 | 3.78 | 3.69 | 3.47 | 3.46 |
| ■ Imagen | 3.16 | 3.78 | 3.75 | 4.12 | 2.88 | 3.53 | 3.94 | 3.75 | 3.61 |
| ■ Flux | 3.31 | 4.31 | 4.28 | 4.12 | 3.5 | 3.94 | 4.44 | 3.75 | 3.96 |
| ■ Imagen-3 | 3.41 | 4.25 | 4.25 | 4.38 | 4.09 | 3.88 | 4.72 | 4.41 | 4.17 |
| ● Custom-Diff | 3.22 | 3.5 | 3.53 | 3.72 | 2.66 | 3.44 | 3.47 | 2.75 | 3.29 |
| ● DreamBooth | 3.16 | 3.66 | 3.31 | 3.88 | 2.94 | 3.53 | 3.41 | 3.09 | 3.37 |
| ● Instruct-Imagen | 2.53 | 3.25 | 2.75 | 2.41 | 1.91 | 2.94 | 3.0 | 2.28 | 2.63 |
| MLLM Entity Alignment | | | | | | | | | |
| ■ SD | 2.34 | 3.12 | 3.09 | 2.28 | 1.22 | 1.97 | 1.69 | 3.44 | 2.39 |
| ■ Imagen | 2.81 | 3.16 | 3.25 | 2.72 | 1.12 | 2.41 | 1.59 | 3.19 | 2.53 |
| ■ Flux | 2.41 | 2.97 | 2.47 | 1.75 | 1.12 | 1.72 | 1.22 | 2.69 | 2.04 |
| ■ Imagen-3 | 3.19 | 3.34 | 4.06 | 3.03 | 1.31 | 2.25 | 1.69 | 3.75 | 2.83 |
| ● Custom-Diff | 2.94 | 3.41 | 1.78 | 3.28 | 1.88 | 3.34 | 2.91 | 2.03 | 2.7 |
| ● DreamBooth | 3.12 | 3.5 | 2.66 | 2.59 | 1.91 | 3.53 | 2.59 | 2.66 | 2.82 |
| ● Instruct-Imagen | 3.88 | 4.06 | 3.91 | 4.03 | 3.22 | 4.09 | 3.31 | 3.25 | 3.72 |

(■: Text-to-Image Models, ●: Retrieval-augmented Models)

whereas Imagen and other models fall below 80. Similarly, Imagen-3 demonstrates strong performance in *Composition*, with a score of 94.6, highlighting its robust instruction-following capabilities in this aspect. In contrast, DreamBooth leads in *Style* with the highest score of 90.2, followed by Custom-Diffusion (84.3) and Stable-Diffusion (80.4), showcasing the strong ability of Stable Diffusion and its retrieval-augmented variants to generate accurate styles. In the *Material* category, all models struggle with instruction adherence, though Imagen-3 achieves a relatively higher score.

Table 4: Detailed breakdown of human evaluation.

| Model | Basic | Location | Composition | Style | Material | Overall |
|---|---|---|---|---|---|---|
| Faithfulness to entity | | | | | | |
| ■ SD | 3.08 | 2.87 | 2.03 | 2.16 | 2.65 | 2.51 |
| ■ Imagen | 3.96 | 3.16 | 2.38 | 2.53 | 2.76 | 2.82 |
| ■ Flux | 3.50 | 3.04 | 2.67 | 2.32 | 2.54 | 2.74 |
| ■ Imagen-3 | 4.08 | 3.55 | 2.78 | 2.79 | 3.13 | 3.17 |
| ● Custom-Diff | 4.34 | 2.77 | 2.73 | 3.12 | 2.80 | 2.90 |
| ● DreamBooth | 4.18 | 3.33 | 2.72 | 2.95 | 2.94 | 3.08 |
| ● Instruct-Imagen | 4.90 | 4.18 | 4.16 | 4.23 | 4.21 | 4.22 |
| Instruction-following | | | | | | |
| ■ SD | 100.0 | 80.0 | 64.3 | 80.4 | 53.1 | 72.2 |
| ■ Imagen | 90.0 | 75.6 | 80.4 | 66.7 | 59.2 | 72.2 |
| ■ Flux | 90.0 | 90.0 | 83.9 | 58.8 | 61.2 | 76.9 |
| ■ Imagen-3 | 90.0 | 87.8 | 94.6 | 76.5 | 69.4 | 83.6 |
| ● Custom-Diff | 100.0 | 72.2 | 66.1 | 84.3 | 46.9 | 69.5 |
| ● DreamBooth | 90.0 | 77.8 | 67.9 | 90.2 | 53.1 | 73.8 |
| ● Instruct-Imagen | 100.0 | 46.7 | 58.9 | 47.1 | 20.4 | 46.5 |

(■: Text-to-Image Models, ●: Retrieval-augmented Models)

# 6 Details of the Automatic Metric Scores Across Prompts

We provide a detailed breakdown of automatic metrics across five evaluation tasks in the KITTEN benchmark, as shown in Tab. 5. We observe that *Location* achieves the highest average image-text score (0.334), followed by *Composition* (0.330), *Style* (0.326), and *Material* (0.319). This pattern suggests that generating entities within a given context is relatively easier, whereas accurately modifying materials remains more challenging. In the categories of *Location*, *Composition*, and *Material*, the base models tend to achieve higher image-text scores, with Imagen-3 attaining the highest score among them. Among retrieval-based models, DreamBooth outperforms Custom Diffusion and Instruct-Imagen. Notably, in *Material*, Imagen scores lower (0.317) than DreamBooth (0.319) and Custom Diffusion (0.320), indicating greater difficulty for Imagen in this category.

For the image-entity score, we find that *Location* again ranks highest with an average score of 0.431, followed by *Material* (0.417), *Style* (0.399), and *Composition* (0.364). This ranking highlights the challenge of maintaining entity fidelity when prompts require complex compositions. In the categories of *Location*, *Composition*, and *Material*, retrieval-augmented models generally outperform base models, with Instruct-Imagen achieving the highest score, followed by DreamBooth and Custom Diffusion. Imagen-3 and Flux also surpass their respective predecessors, Imagen and Stable-Diffusion. However, in an unexpected result for *Composition*, Imagen (0.333) outperforms Imagen-3 (0.317).

While most categories exhibit higher image-entity scores for *Material* compared to *Style*, Imagen-3 and Flux surprisingly perform better on styles. Additionally, other models tend to achieve higher image-entity scores on *Style* than on *Composition*, with DreamBooth being an exception. These results highlight the varying strengths of different models in preserving entity fidelity across diverse prompt types.

Table 5: Detailed breakdown of automatic metrics.

| Model | Basic | Location | Composition | Style | Material | Overall |
|---|---|---|---|---|---|---|
| Image-Text Alignment: CLIP-T | | | | | | |
| ■ SD | 0.308 | 0.342 | 0.333 | 0.346 | 0.332 | 0.338 |
| ■ Imagen | 0.304 | 0.338 | 0.329 | 0.328 | 0.317 | 0.329 |
| ■ Flux | 0.299 | 0.340 | 0.340 | 0.308 | 0.324 | 0.329 |
| ■ Imagen-3 | 0.303 | 0.347 | 0.345 | 0.322 | 0.333 | 0.338 |
| ● Custom-Diff | 0.308 | 0.326 | 0.324 | 0.335 | 0.320 | 0.324 |
| ● DreamBooth | 0.308 | 0.336 | 0.327 | 0.341 | 0.319 | 0.331 |
| ● Instruct-Imagen | 0.302 | 0.312 | 0.311 | 0.305 | 0.291 | 0.307 |
| Image-Entity Alignment: DINO | | | | | | |
| ■ SD | 0.467 | 0.379 | 0.295 | 0.311 | 0.349 | 0.350 |
| ■ Imagen | 0.472 | 0.409 | 0.333 | 0.376 | 0.385 | 0.386 |
| ■ Flux | 0.464 | 0.394 | 0.316 | 0.383 | 0.368 | 0.380 |
| ■ Imagen-3 | 0.483 | 0.412 | 0.317 | 0.403 | 0.391 | 0.389 |
| ● Custom-Diff | 0.578 | 0.411 | 0.348 | 0.359 | 0.400 | 0.388 |
| ● DreamBooth | 0.597 | 0.440 | 0.376 | 0.371 | 0.417 | 0.412 |
| ● Instruct-Imagen | 0.626 | 0.573 | 0.566 | 0.593 | 0.606 | 0.582 |

(■: Text-to-Image Models, ●: Retrieval-augmented Models)

# 7 Details of the MLLM Evaluation Across Prompts

We provide a detailed breakdown of MLLM evaluation results across five evaluation tasks in the KITTEN benchmark, as shown in Tab. 6. For MLLM text alignment, the ranking across prompt types largely mirrors the overall trend. Imagen-3 consistently achieves the highest overall score (4.17), followed by Flux (3.96) and Imagen (3.61). Among retrieval-augmented models, DreamBooth and Custom-Diffusion perform moderately (3.37 and 3.29, respectively), while Instruct-Imagen scores the lowest (2.63). Imagen-3 leads across most prompt types, particularly excelling in *Composition* (4.59) and *Material* (3.78). Flux also performs well,

especially in *Location* (4.31) and *Composition* (4.23). In contrast, Custom-Diffusion and DreamBooth show weaker alignment, particularly in *Composition* and *Material*, with scores below 3.5. The MLLM entity alignment scores follow a similar pattern, with Instruct-Imagen achieving the highest overall score (3.72), outperforming all other models across all prompt types. It particularly excels in *Composition* (3.91), *Style* (3.67), and *Material* (3.65), demonstrating strong grounding of entity representations in the image. Among the base models, Imagen-3 performs best (2.83), while Flux consistently underperforms across all prompts. These results further underscore the effectiveness of retrieval augmentation in enhancing entity alignment, especially for complex instructions involving *Style* and *Material*.

Table 6: Detailed breakdown of MLLM evaluation.

| Model | Basic | Location | Composition | Style | Material | Overall |
|---|---|---|---|---|---|---|
| MLLM Text Alignment | | | | | | |
| ■ SD | 4.6 | 3.61 | 3.3 | 3.86 | 2.69 | 3.46 |
| ■ Imagen | 5.0 | 3.7 | 3.86 | 3.45 | 3.06 | 3.61 |
| ■ Flux | 5.0 | 4.31 | 4.23 | 3.35 | 3.41 | 3.96 |
| ■ Imagen-3 | 5.0 | 4.28 | 4.59 | 3.75 | 3.78 | 4.17 |
| ● Custom-Diff | 4.6 | 3.3 | 3.2 | 3.84 | 2.51 | 3.29 |
| ● DreamBooth | 4.7 | 3.34 | 3.25 | 3.94 | 2.69 | 3.37 |
| ● Instruct-Imagen | 4.6 | 2.6 | 2.86 | 2.82 | 1.84 | 2.63 |
| MLLM Entity Alignment | | | | | | |
| ■ SD | 2.9 | 2.87 | 2.16 | 1.84 | 2.27 | 2.39 |
| ■ Imagen | 3.2 | 2.89 | 2.48 | 1.94 | 2.41 | 2.53 |
| ■ Flux | 2.0 | 2.24 | 2.43 | 1.71 | 1.59 | 2.04 |
| ■ Imagen-3 | 3.82 | 3.18 | 2.63 | 2.41 | 2.63 | 2.83 |
| ● Custom-Diff | 3.7 | 2.7 | 2.57 | 2.8 | 2.51 | 2.7 |
| ● DreamBooth | 4.1 | 3.09 | 2.55 | 2.45 | 2.76 | 2.82 |
| ● Instruct-Imagen | 4.2 | 3.61 | 3.91 | 3.67 | 3.65 | 3.72 |

(■: Text-to-Image Models, ●: Retrieval-augmented Models)

# 8 Details of the Correlation with Human Evaluation

To quantify the consistency between automatic metrics and human evaluation results, we computed Pearson and Spearman correlations. CLIP-T and CLIP-I show moderate alignment with user evaluations. Here, we provide a detailed breakdown of the correlations across eight domains in Tab. 7. While these metrics show some alignment with human evaluations, discrepancies remain in certain categories. Notably, the correlation between CLIP-T and the instruction-following score in the landmark domain, as well as the correlation between CLIP-I and the faithfulness score in the plant domain, is negative. This underscores the limitations of automatic metrics in fully capturing human judgment. Although DINO demonstrates stronger alignment with user evaluations of faithfulness, correlation variability persists across categories. For instance, correlations are lower in the vehicle and landmark domains, highlighting the need for more accurate automatic metrics to better reflect human evaluations and assess model performance.

Additionally, both MLLM entity alignment and text alignment scores show higher correlations with human evaluations in most domains, indicating stronger consistency as automatic proxies. However, exceptions arise in the aircraft and vehicle domains, where the correlation values are noticeably lower. This suggests that while MLLM-based scores generally align well with human judgments, their effectiveness may be reduced in domains characterized by complex or diverse visual features, such as aircraft and vehicles. Further refinement of these metrics could enhance their robustness and reliability across all categories.

Table 7: Per-category correlation with human evaluation.

| Metric | Type | Aircraft | Vehicle | Cuisine | Flower | Insect | Landmark | Plant | Average |
|---|---|---|---|---|---|---|---|---|---|
| CLIP-T | Pearson | **0.138** | 0.499 | 0.105 | 0.788 | 0.567 | **-0.299** | 0.564 | 0.337 |
| | Spearman | **0.168** | 0.454 | 0.286 | 0.836 | 0.554 | **-0.166** | 0.558 | 0.384 |
| CLIP-I | Pearson | 0.330 | 0.195 | **0.100** | 0.215 | 0.528 | 0.356 | **-0.051** | 0.239 |
| | Spearman | 0.548 | 0.323 | **0.238** | 0.287 | 0.503 | 0.623 | **-0.143** | 0.340 |
| DINO | Pearson | 0.655 | **0.367** | 0.549 | 0.551 | **0.277** | 0.680 | 0.492 | 0.510 |
| | Spearman | 0.575 | **0.311** | 0.735 | 0.430 | **0.210** | 0.700 | 0.565 | 0.504 |
| MLLM-T | Pearson | 0.477 | 0.687 | 0.583 | 0.673 | 0.586 | 0.790 | 0.575 | 0.618 |
| | Spearman | 0.478 | 0.605 | 0.536 | 0.604 | 0.577 | 0.748 | 0.510 | 0.589 |
| MLLM-I | Pearson | 0.468 | 0.467 | 0.754 | 0.783 | 0.746 | 0.742 | 0.705 | 0.703 |
| | Spearman | 0.475 | 0.393 | 0.711 | 0.764 | 0.686 | 0.748 | 0.704 | 0.695 |

## 9  Dataset Statistics

The KITTEN benchmark focuses on evaluating faithfulness to knowledge-grounded concepts. To ensure diversity, we select entities from eight specialized domains and construct diverse prompts in each domain for evaluation. For each entity, we collect a set of support images as inputs to assess retrieval-augmented models, where support images are used to enhance the model's predictions. In addition, we collect a set of evaluation images for conducting human evaluation. A detailed breakdown of the data statistics is provided in Tab. 8.

Next, we present the distribution of different evaluation tasks in Tab. 9. The types of evaluation tasks include: 1) generating the knowledge entity, 2) placing the knowledge entity in context, 3) composing multiple entities, 4) creation in different styles, and 5) creation using different materials.

Table 8: Statistics of KITTEN benchmark.

| Domain | #Entities | #Prompts | #Support Images | #Eval Images | # (Entitiy, Prompt) |
|---|---|---|---|---|---|
| Aircraft | 48 | 20 | 469 | 237 | 960 |
| Vehicle | 50 | 20 | 500 | 250 | 1000 |
| Flower | 18 | 20 | 180 | 90 | 360 |
| Insect | 50 | 20 | 500 | 250 | 1000 |
| Plant | 48 | 20 | 480 | 240 | 960 |
| Landmark | 50 | 20 | 500 | 250 | 1000 |
| Cuisine | 31 | 20 | 310 | 155 | 620 |
| Sport | 27 | 20 | 270 | 135 | 540 |

Table 9: Statistics of KITTEN evaluation tasks.

| Evaluation task | #Prompts | Percentage (%) |
|---|---|---|
| Basic | 295 | 4.58 |
| Location | 1969 | 30.57 |
| Composition | 1467 | 22.78 |
| Style | 1365 | 21.20 |
| Material | 1344 | 20.87 |

## 10  Details on Human Evaluation

We employ five annotators per image to ensure robust assessments. The raters are hired through Prolific.com, a third-party rating service. For the binary task ("Adherence to Prompt Beyond References"), we observe

agreement among at least 4 out of 5 annotators in 75% of cases and perfect agreement (5 out of 5) in 44% of cases. For the Likert scale task ("Faithfulness to Reference Entity"), we calculate Krippendorff's Alpha of 0.60, indicating good agreement for a subjective task of this complexity. Additionally, we achieve an IoU-like score of 0.53, which penalizes outliers and demonstrates moderate consensus, and an average pairwise Cohen's Kappa of 0.25, reflecting fair pairwise agreement. The average standard deviation of ratings is 0.74, reflecting moderate variability in annotator judgments. These metrics collectively demonstrate the reliability of our human evaluation.

## 11 Details on Evaluated Models

To facilitate the reproducibility of the KITTEN benchmark experiments, we provide detailed descriptions of the inference setups for all evaluated models. All models generate a single sample for each prompt without any selection or reranking. For backbone text-to-image models (DALL·E-2, SD-1.5, SD-2.1, SD-3, SD-XL, Imagen, Flux, Imagen-3) and unified models (Show-o, Janus-Pro, Emu3), we follow their released inference configurations, where no temperature parameter is used. These models do not require fine-tuning and do not take support images as inputs; they directly generate an image from the input prompt. For retrieval-augmented models, the in-context learning methods (BLIP-Diffusion, IP-Adapter, Instruct-Imagen) also do not require fine-tuning. They take a single support image together with the text prompt as input; we randomly select the support image. For fine-tuning-based methods (Custom-Diff, DreamBooth), we fine-tune the model for each entity using 10 reference images that are disjoint from the evaluation set. We use the AdamW optimizer with a learning rate of $5 \times 10^{-6}$ and default $\beta$ values ($\beta_1 = 0.9$, $\beta_2 = 0.999$). Training is conducted for 1,000 steps with a batch size of 5. After fine-tuning, these models take only the text prompt as input to generate an image. All images are generated using evaluation text prompts, with no overlap in prompts or images between the test and support sets. Evaluation with GPT-4o-mini is performed via API calls. Experiments are conducted on a cluster of eight NVIDIA A100 GPUs (each with 40GB of memory). Fine-tuning takes approximately 20 minutes per entity, and inference requires roughly 5 seconds per image.

## 12 Human Annotation Instructions

We provide the complete instructions given to human raters for evaluating the generated images in the KITTEN benchmark. We note that our human evaluation does not require raters to strictly match the reference image. Instead, they are explicitly instructed to conduct their own research (e.g., via Google Search) to verify the entity's appearance and assess whether the generated image accurately depicts it. This allows for acceptable variations, such as different flower colors or alternative representations of the same landmark.

### Rater Instructions

In this task, you will be provided with a Prompt, Reference Images, and a Generated Image. Your task is to assess the factual accuracy of the generated image with respect to the prompt and the reference images. The goal is to ensure that the entity described in the prompt is factually correct and accurately represented. While the reference images offer a visual starting point, you may conduct your own research (e.g., Google Search) to clarify the appearance of the entity and ensure its accurate depiction in the generated image.

**Part 1: Reference Alignment**

**Faithfulness to Prompt Entity (Factuality)**

Your first task is to evaluate how faithfully the generated image represents the reference entity. Consider whether the reference entity's key features and overall appearance are accurately depicted.

*Question:* How faithfully does the generated image represent the entity mentioned in the prompt?

*Candidate Answers:*
1 (Not faithful at all): The generated image does not represent the reference entity at all. There are no discernible visual similarities to the reference entity.
2 (Barely faithful): The generated image faintly represents the reference entity, with significant effort needed

to see any resemblance. Minor visual elements may be present, but crucial features or characteristics are missing or significantly misrepresented.

3 (Somewhat faithful): The generated image somewhat represents the reference entity, but it is not prominent. There is a clear visual connection in terms of composition, style, or some key elements, but there are noticeable differences, omissions, or misinterpretations.

4 (Mostly faithful): The generated image mostly represents the reference entity and clearly presents it. It draws strong visual inspiration with a strong connection in terms of overall composition, style, key elements, and/or subject matter, despite some variations in details.

5 (Completely faithful): The generated image fully represents the reference entity accurately. It captures all key elements, composition, and style in a way that is almost identical to the reference entity.

### Open Questions for Reference Alignment

*Visual Similarities:* Describe any visual similarities between the generated image and the reference images, focusing on elements that enhance the recognizability of the entity. Be specific about shape, color, texture, composition, objects, or overall style.

*Visual Differences:* Describe any differences in the generated image that negatively impact its faithfulness to the entity in the prompt and are not specified by the prompt. Focus on aspects that affect recognizability, and avoid mentioning changes that do not impact identification (e.g., angle or color for cars or aircraft).

### Part 2: Text-Image Adherence

### Adherence to Prompt Beyond References

Next, evaluate whether the generated image accurately and comprehensively depicts all aspects of the scene or entity described in the prompt that are not already reflected in the reference images. This involves checking for details in the prompt that go beyond what is shown in the reference images.

*Question:* Does the generated image accurately and comprehensively depict any aspects of the scene or entity described in the prompt that are not already reflected in the reference images?

*Candidate Answers:*

Yes: The generated image accurately and comprehensively depicts aspects of the scene or entity described in the prompt that are not already reflected in the reference images.

No: The generated image fails to accurately and comprehensively depict aspects of the scene or entity described in the prompt that are not already reflected in the reference images.

*If the answer is No:*

*Misalignments:* Explain the misalignments between the generated image and the prompt text. Focus on elements or concepts that are not present in the reference images. Be specific about which aspects are missing, inaccurate, or misrepresented.

### Optional: Open-Ended Feedback

*Question:* Do you have any other comments or observations about the generated image? (optional)

## 13   MLLM Annotation Instructions

We provide the complete instructions given to GPT-4o-mini for evaluating the generated images in the KITTEN benchmark.

### Rater Instructions

Your task is to assess the factual accuracy of the generated image with respect to the prompt and the reference images. The goal is to ensure that the entity described in the prompt is factually correct and accurately represented.

**Part 1:**

In this task, you will be provided with a Prompt, Reference Images, and a Generated Image. Evaluate how faithfully the generated image represents the reference entity. Consider whether the key features and overall appearance of the reference entity are accurately depicted.

*Question 1*: How faithfully does the generated image represent the entity mentioned in the prompt?

*Candidate Answers*:
1 (Not faithful at all): The generated image does not represent the reference entity at all. There are no discernible visual similarities to the reference entity.
2 (Barely faithful): The generated image faintly represents the reference entity, with significant effort needed to see any resemblance. Minor visual elements may be present, but crucial features or characteristics are missing or significantly misrepresented.
3 (Somewhat faithful): The generated image somewhat represents the reference entity, but it is not prominent. There is a clear visual connection in terms of composition, style, or some key elements, but there are noticeable differences, omissions, or misinterpretations.
4 (Mostly faithful): The generated image mostly represents the reference entity and clearly presents it. It draws strong visual inspiration with a strong connection in terms of overall composition, style, key elements, and/or subject matter, despite some variations in details.
5 (Completely faithful): The generated image fully represents the reference entity accurately. It captures all key elements, composition, and style in a way that is almost identical to the reference entity.

*Answer in the exact format below*:
Question 1:
Answer: [1–5]
Reason: [Provide a clear explanation for your answer]

**Part 2:**

In this task, you will be provided with a Prompt and a Generated Image. Evaluate how well the generated image captures all aspects described in the prompt. Focus on background elements, contextual details, materials, styles, and other visual features.

*Question 2*: How well does the generated image depict the details described in the prompt?

*Candidate Answers*:
1 (Not at all): None of the described elements are present in the image.
2 (Slightly): A few minor elements are present, but most are missing or inaccurate.
3 (Moderately): Some elements are present and somewhat accurate, but others are missing or misrepresented.
4 (Mostly): Most of the described elements are clearly and accurately depicted.
5 (Completely): All relevant aspects of the prompt are thoroughly and accurately represented.

*Answer in the exact format below*:
Question 2:
Answer: [1–5]
Reason: [Provide a clear explanation for your answer]

## 14   Limitations

We discuss the limitations of multi-entity prompts, complex reasoning prompts, and choices of Wikipedia entities below.

**Multi-entity prompts.**   KITTEN focuses on single-entity prompts because our goal is to stress-test a model's ability to reconstruct visual entities from diverse domains under a basic, controlled setup. Multi-entity prompts pose additional challenges, as the model must preserve each entity's identity without mixing or confusion, a task that is not unique to specialized entities and can be equally difficult for common objects. Although multi-entity generation is more complex, retrieval-augmented models such as Custom Diffusion have shown that, under the same fine-tuning procedure, a model can handle multiple target entities with

quality comparable to single-entity generation. Therefore, we believe that our findings can be extended to multi-entity settings. In particular, the trade-off we observe between maintaining high entity fidelity and following creative prompt instructions is likely to persist. Our proposed future direction, combining a strong backbone with a well-designed retrieval-augmented approach, should also hold in multi-entity scenarios.

**Complex reasoning prompts.** KITTEN does not include prompts that require complex reasoning. We observe that several concurrent benchmarks evaluate models using complex prompts involving implicit reasoning (e.g., WISE, WorldGenBench, R2I-Bench, OmniGenBench, ABP), but none focus on generating a broad set of real-world entities across diverse domains, which is the primary focus of KITTEN. As noted in the WISE paper, many complex reasoning prompts (e.g., "the tallest building in Manhattan") can be reformulated into entity-centered prompts through prompt rewriting, which consistently improves model scores. Consistent with this observation, our analysis intentionally abstracts away the "language reasoning" component and instead focuses on the core challenge of "visual entity generation" using explicitly instructed prompts that effectively represent the rewritten versions.

**Choices of Wikipedia entities.** While KITTEN prompts are currently limited to Wikipedia and eight domains, it covers the largest number of visual entities among existing benchmarks, including 322 entities across 8 domains and 6,440 prompts. KITTEN provides broader coverage than alternative image customization benchmarks, which focus on different tasks and include fewer entities and prompts. For example, Dream-Bench (Ruiz et al., 2023), DreamBench-v2 (Chen et al., 2023), CustomConcept101 (Kumari et al., 2023), DreamBench++ (Peng et al., 2025), MM-Diff (Wei et al., 2024), and StoryMaker (Zhou et al., 2024) evaluate only 30, 30, 101, 150, 25, and 40 subjects, respectively, with 750, 220, 2020, 1350, 500, and 800 prompts. We emphasize that this work represents an initial step toward building a large-scale evaluation dataset for the novel task of generating real-world visual entities, and we plan to expand its scope in future work. This includes incorporating domains not yet covered, such as the VQA data in OVEN-Wiki, as well as extending the dataset by selecting visually relevant categories and entities from resources like OntoNotes and WordNet.

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

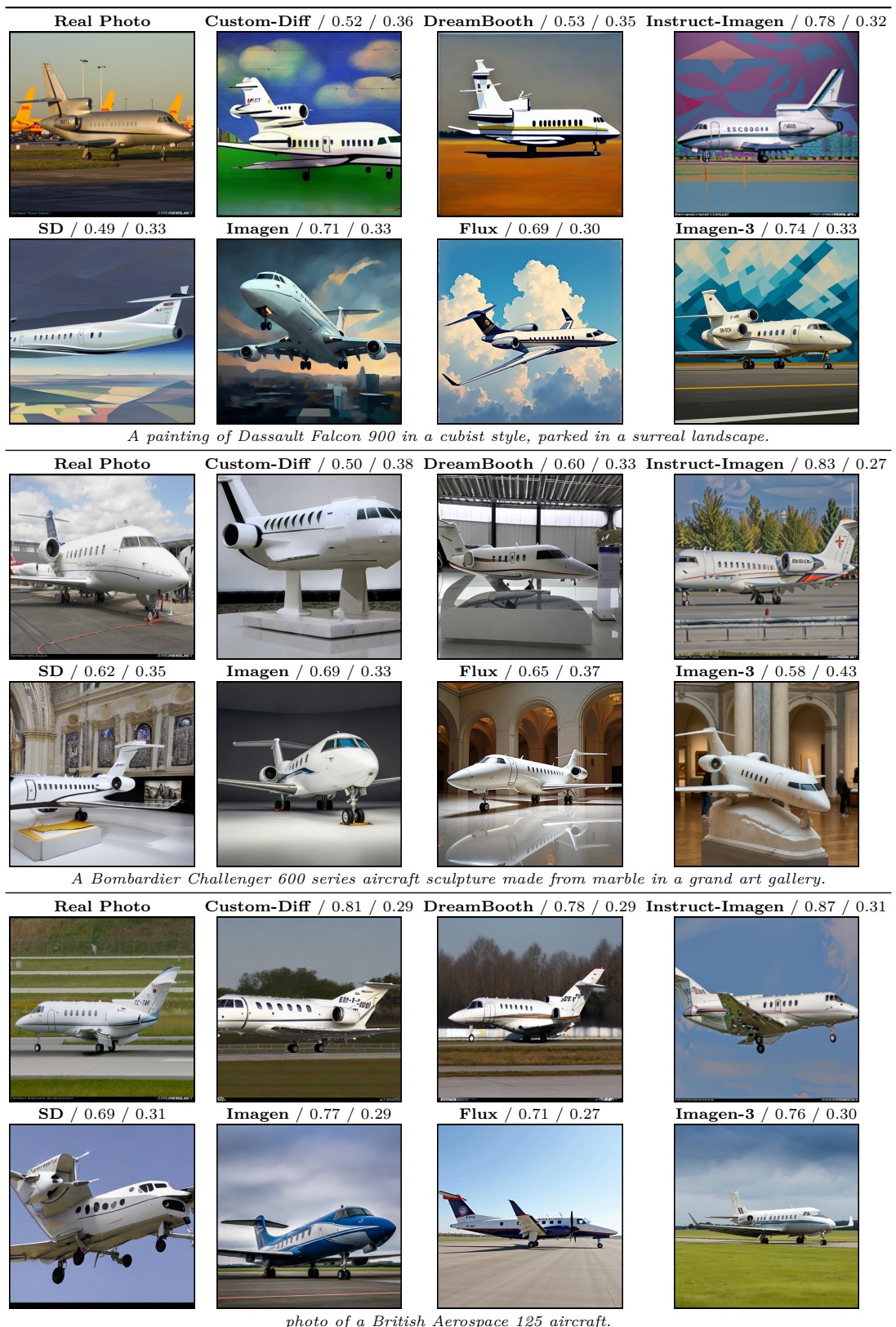

Figure 1: **Qualitative results** for the aircraft domain, including the DINO and CLIP-T scores.

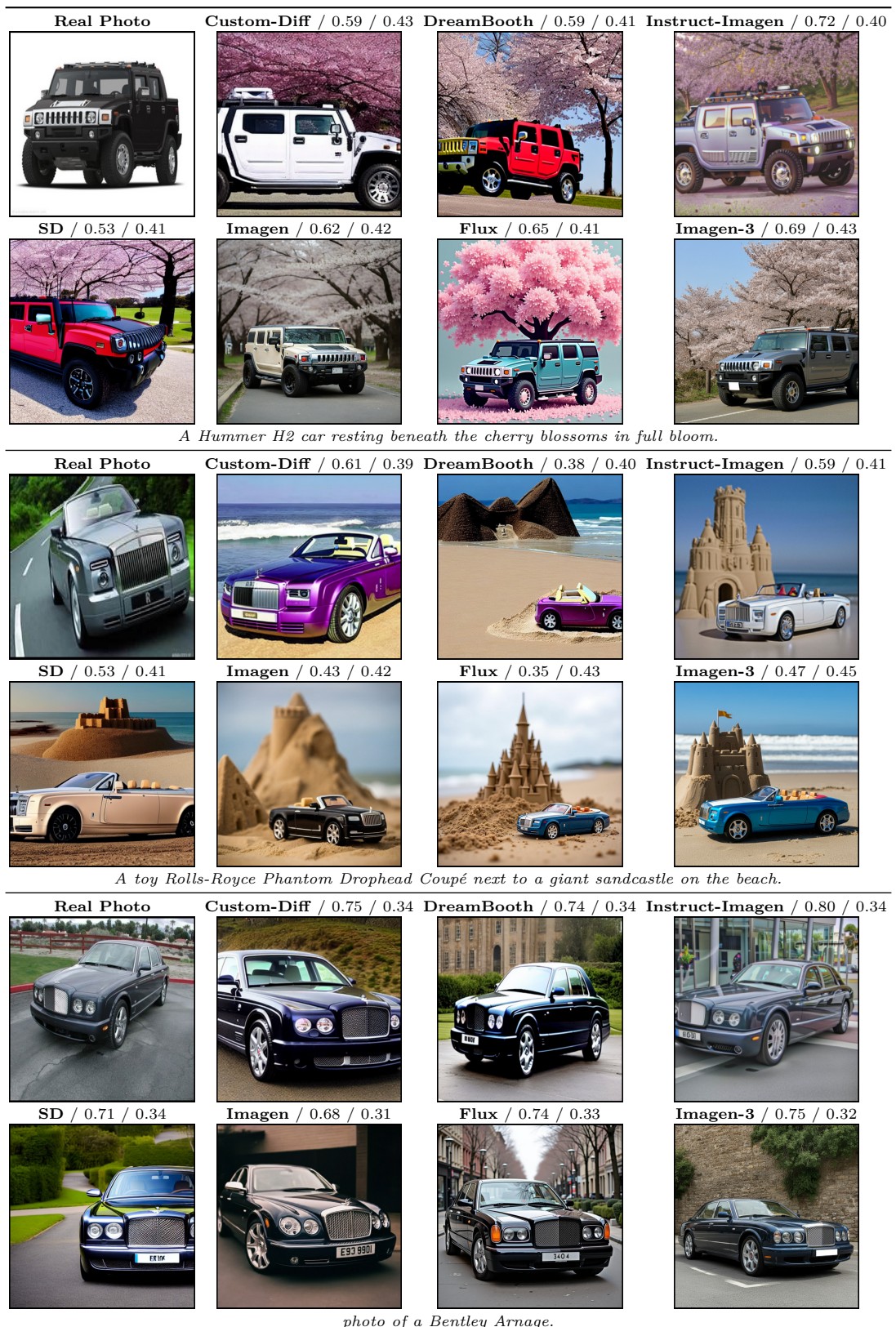

Figure 2: **Qualitative results** for the vehicle domain, including the DINO and CLIP-T scores.

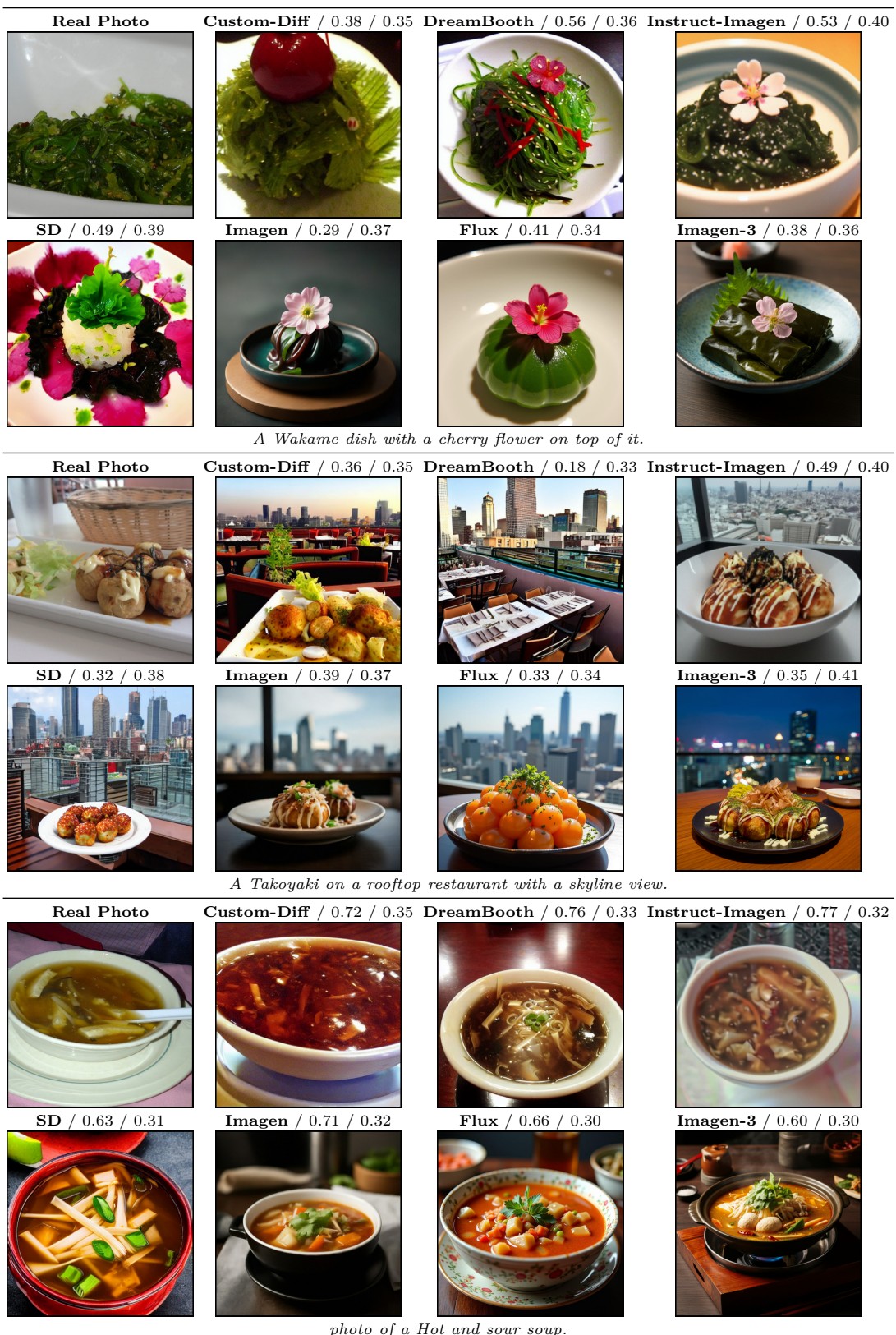

Figure 3: **Qualitative results** for the cuisine domain, including the DINO and CLIP-T scores.

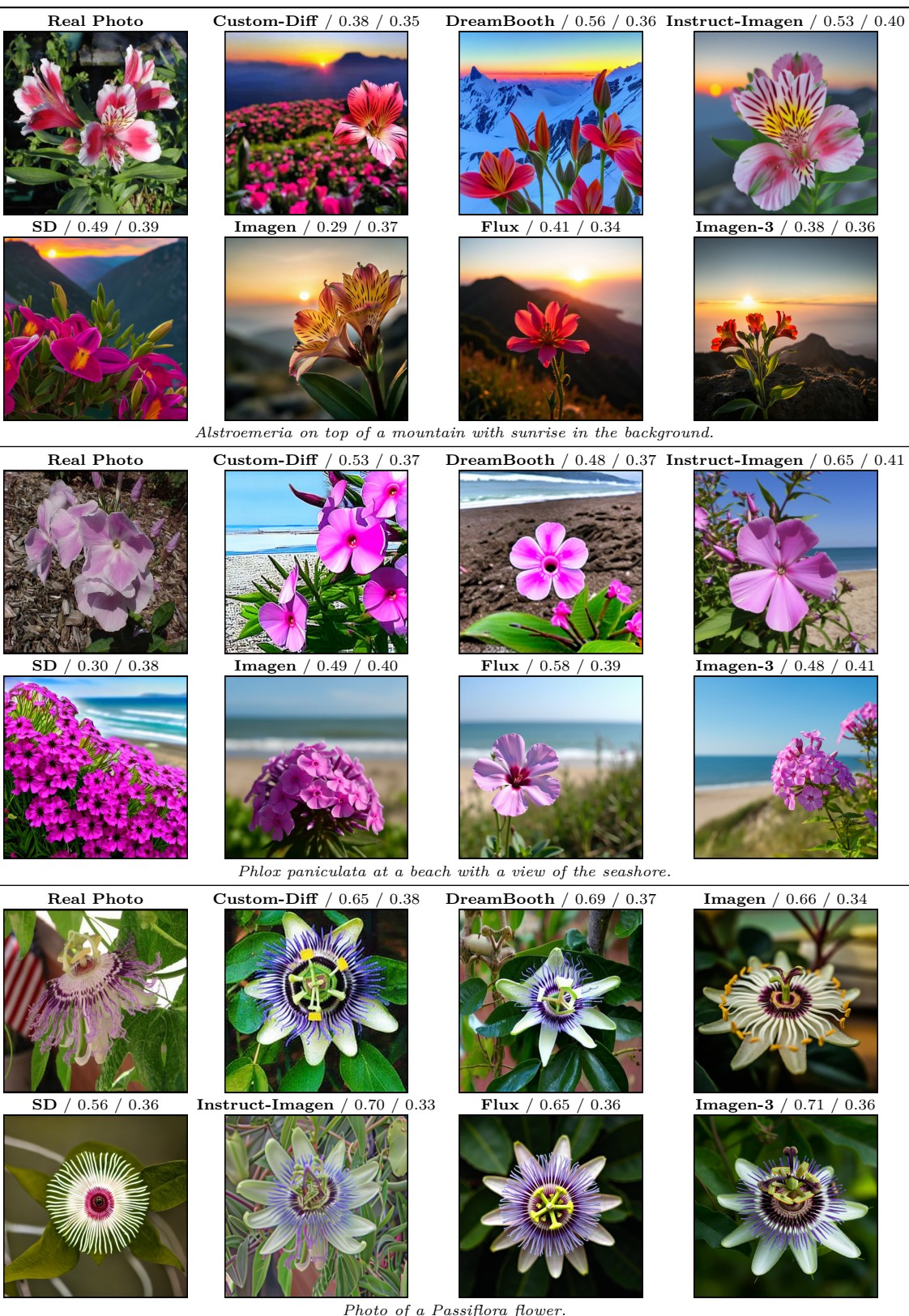

Figure 4: **Qualitative results** for the flower domain, including the DINO and CLIP-T scores.

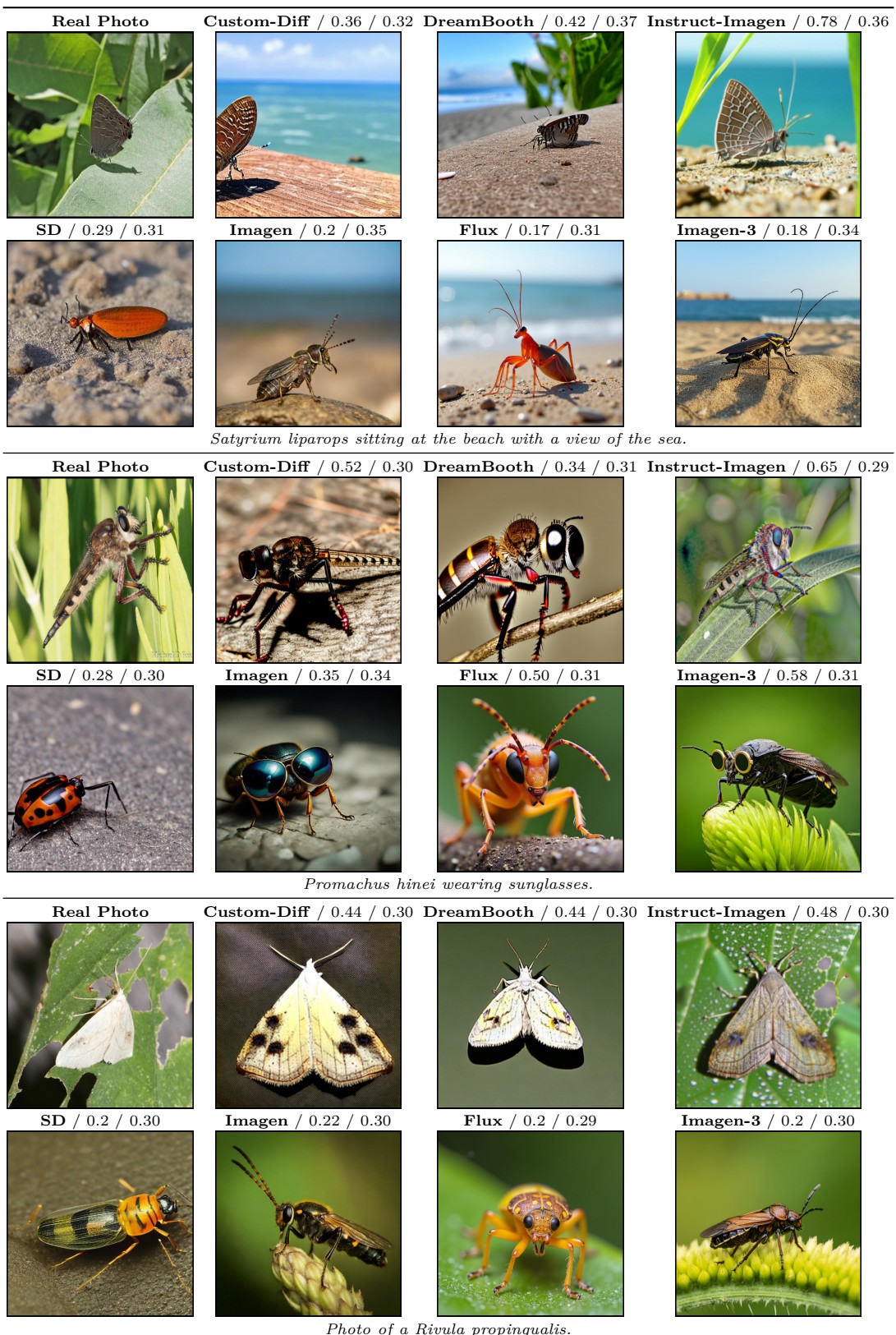

Figure 5: **Qualitative results** for the insect domain, including the DINO and CLIP-T scores.

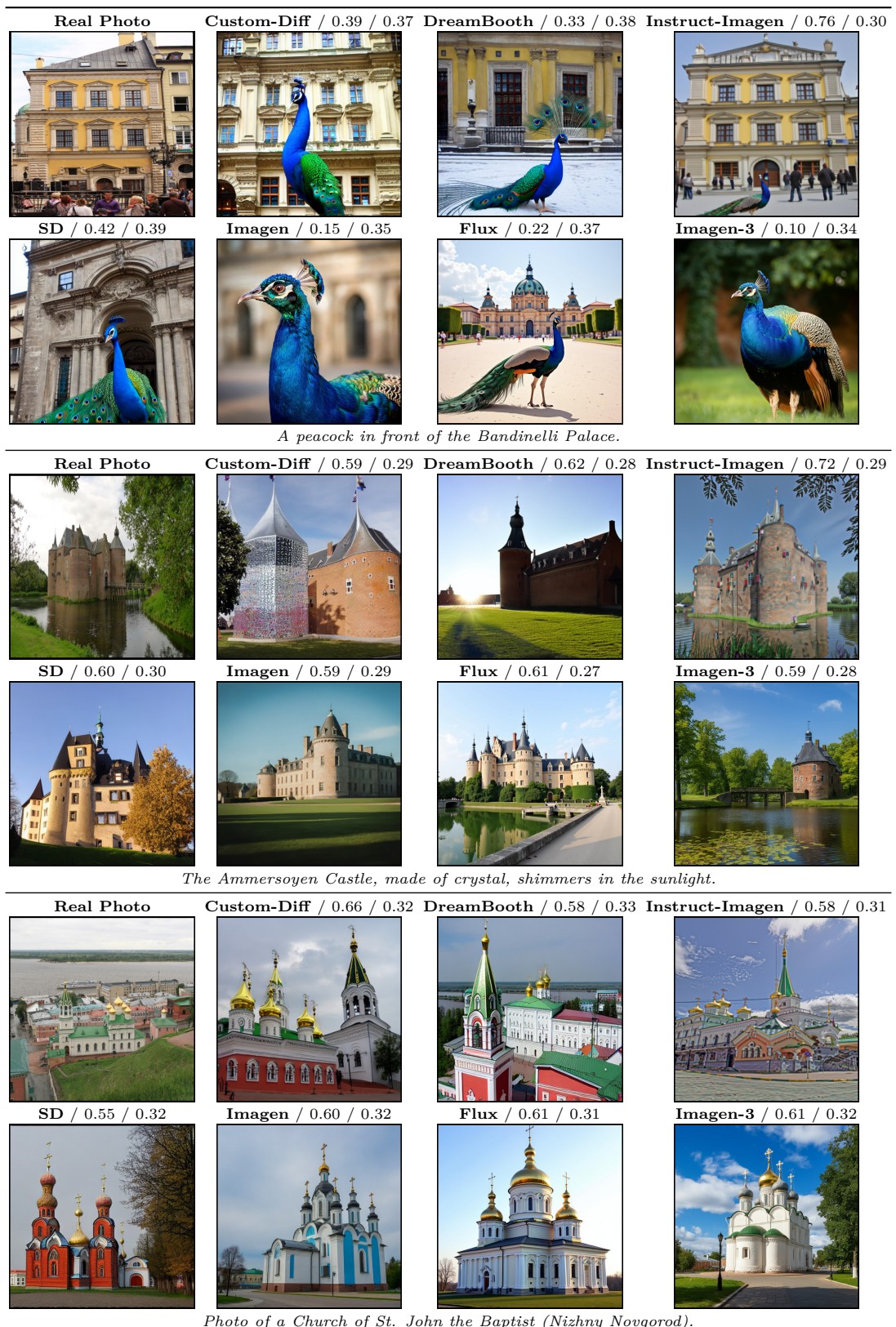

Figure 6: **Qualitative results** for the landmark domain, including the DINO and CLIP-T scores.

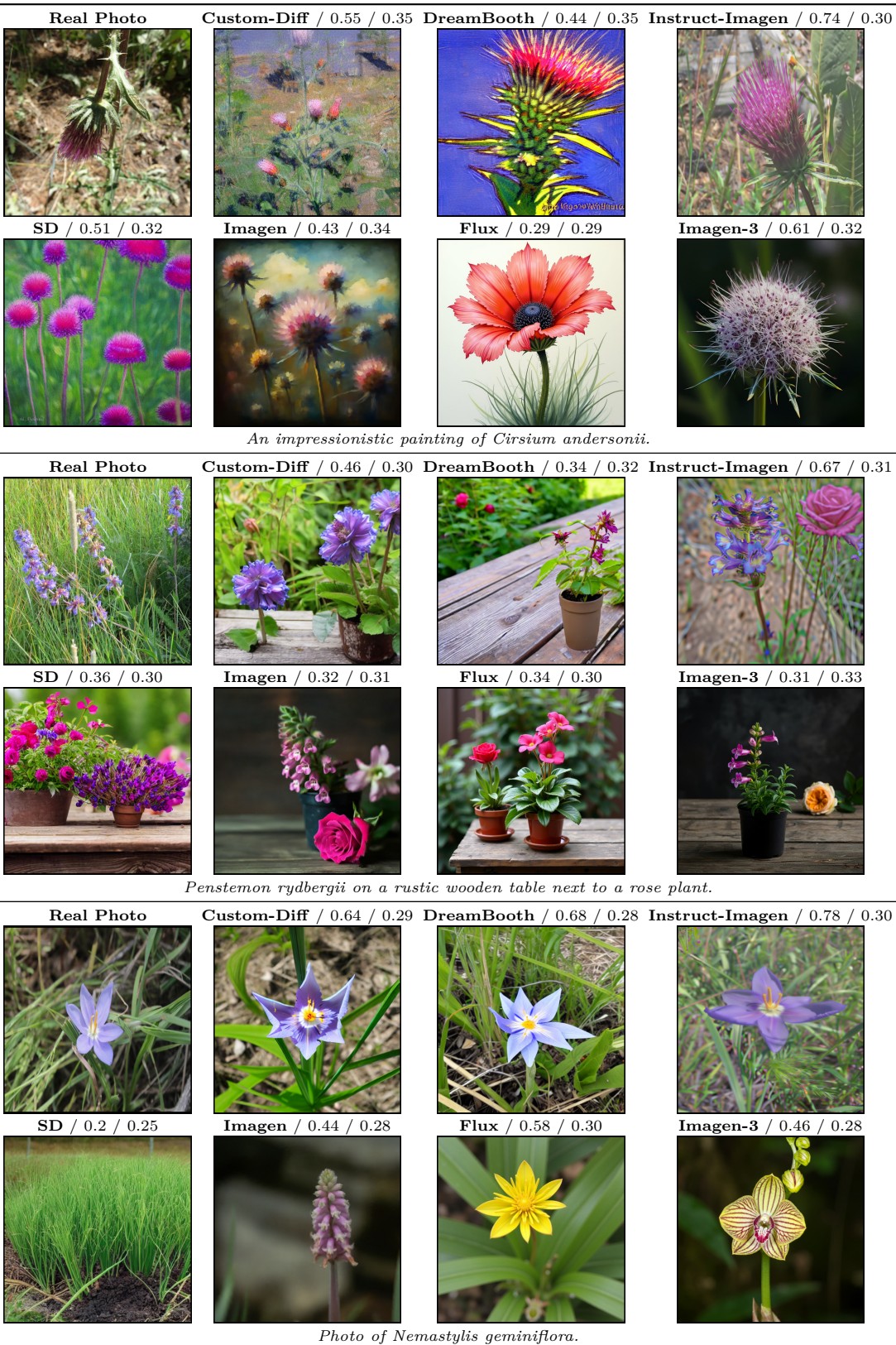

Figure 7: **Qualitative results** for the plant domain, including the DINO and CLIP-T scores.

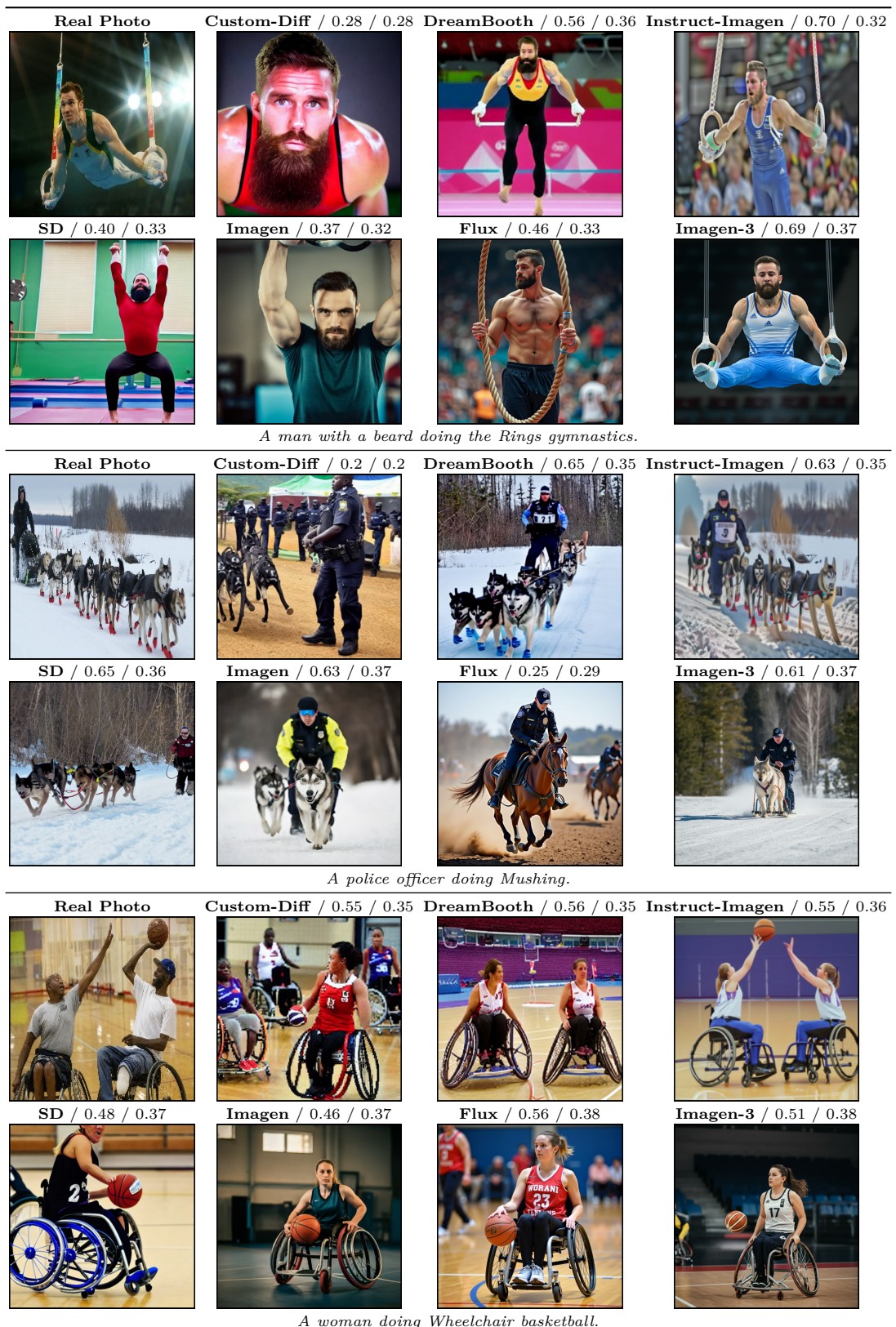

Figure 8: **Qualitative results** for the sport domain, including the DINO and CLIP-T scores.