# OpenReview forum: "KITTEN: A Knowledge-Integrated Evaluation of Image Generation on Visual Entities"
_TMLR — Accepted by TMLR_

### Review · Reviewer_tQZs · 2025-11-23

**Summary Of Contributions:**

This paper proposes a benchmark dataset and evaluation suite KITTEN for assessing whether text-to-image models can accurately generate real-world visual entities grounded in a knowledge source Wikipedia.

strengths
1. The paper builds a focused benchmark that directly checks whether a model can draw a specific real-world entity (e.g., a particular palace) correctly, not just make a pretty picture of generic scene.

2. The paper compares backbone generators, retrieval-augmented methods with reference images, and unified multimodal models, using human judgment plus automatic and MLLM-based metrics.

Weaknesses
1. KITTEN's samples may overlap with the training data
As mentioned in paper "Kitten uses prompts derived from visual entities documented in Wikipedia", it may cause overlap with the training data. Many models are trained on large web datasets which likely include the same or similar Wikipedia images, also similar captions.


2. It's not actually "Knowledge-Intensive" benchmark

The paper’s title and motivation repeatedly say KITTEN is a benchmark for “Knowledge-Intensive” image generation on real-word Entitis. A benchmark can be called knowledge-intensive if it use stored world knowledge that is not directly written in the input prompt.

KITTEN's prompts give the entity name explicitly intead of world knowledge such as facts, relations, causal reasoning (e.g., “the tallest building in Manhattan”). The paper also acknowledges the knowledge-based reasoning is not available in current straightforward, entity-focused prompts (see section 6).

The KITTEN benchmark checks given the entity name if the generated image visually matches the prompt which takes the form of modifier words+entity name, e.g. "Photo of Bandinelli Palace.", "An oil painting of Bandinelli Palace.", "Bandinelli Palace made of crystal."

3. auto metrics does not match human evaluation

The paper relies on CLIP, DINO, and MLLM-based metrics (GPT-4o, Qwen-2.5) to approximate human evaluation, but these metrics sometimes disagree with human rankings or only show moderate correlation with human evaluation. In Table 3, correlation of automatic and MLLM metrics (GPT-4o, Qwen-2.5, DINO, CLIP-I) with human evaluation shows only GPT-4o is moderately correlated with humans, with a pearson correlation of around 0.7, while the others are only around 0.5, and CLIP is particularly weak at around 0.2.

4. Not provide a robust and open automatic metric

The strongest correlation is GPT-4o which is proprietary. The proprietary metric blocks reproducibility and may change over time. As a result, KITTEN does not provide a robust and open automatic metric can replace human annotation.

5. Human evaluation still requires reliability analysis

We don’t know how consistent the ratings are as the agreement statistics (such as Variance or confidence intervals across raters, Cohen's kappa and Krippendorff's alpha for inter-rater reliability) are not reported.

For example, a moderate final average score may hide a strong disagreement, so the variance is needed to show how widely opinions differ.

**Audience:**

Yes

**Audience Explanation:**

The paper proposes a new benchmark for knowledge-intensive text-to-image generation and evaluates multiple models to give insights about visual-world knowledge and evaluation metrics. These topics are relevant to readers working on benchmarking, generative models and multimodal learning.

**Claims And Evidence:**

No

**Claims Explanation:**

See Weaknesses 3. auto metrics does not match human evaluation, 4. Not provide a robust and open automatic metric, 5. Human evaluation still requires reliability analysis above

**Requested Changes:**

See Weaknesses 1 to 5

---

> ### Author Response · Authors · 2025-12-12
>
> ### **Q1. KITTEN's samples may overlap with the training data**
>
> It is true that the images and captions in KITTEN may overlap with the training data of the evaluated models. However, this is not a weakness of the benchmark. Our goal is not to test zero-shot generation, where a model must produce entities it has never seen before. Instead, KITTEN is designed to evaluate whether a model can faithfully generate real-world entities that do appear in the training data.
>
> The benchmark examines both high-frequency and low-frequency entities. If an entity appears often in the training data, a strong model should reliably generate accurate and recognizable images of it. For long-tail entities that appear only a few times, KITTEN tests whether the model can still represent them, revealing how well the training data covers rare cases and whether the model generalizes beyond dominant categories.
>
> ---
>
> ### **Q2. “Knowledge-Intensive” benchmark**
>
> We will revise the term “knowledge-intensive” in the paper. Both directly incorporating world knowledge through explicit terms and evaluating it via implicit reasoning are valid approaches to assessing a model’s knowledge. In the title and motivation, we emphasize that KITTEN focuses on the generation of visual entities in an explicit scenario.
>
> It is not a limitation that KITTEN’s prompts are entity-focused rather than reasoning-focused. Several concurrent benchmarks (e.g., WISE, WorldGenBench, R2I-Bench, OmniGenBench, ABP) evaluate models using complex prompts involving implicit reasoning, but none focus on generating a broad set of real-world entities across diverse domains.
>
> KITTEN fills an important gap compared to concurrent benchmarks by emphasizing the reconstruction of detailed visual features (e.g., shape and color) of nameable visual entities. We explicitly collect evaluation images for target entities and directly compare them with generated outputs, a feature not addressed by other benchmarks. Additionally, KITTEN evaluates retrieval-augmented models, enabling us to examine whether explicitly providing reference images improves entity generation, another aspect missing from other work.
>
> As noted in the WISE paper, many complex reasoning prompts (e.g., “the tallest building in Manhattan”) can be reformulated into entity-centered prompts through rewriting, which consistently improves model scores. Consistent with this observation, our analysis abstracts away the “language reasoning” component and focuses on the core challenge of “visual entity generation” using explicitly instructed prompts.
>
> ---
>
> ### **Q3. Automatic metrics do not always match human evaluation**
>
> We provide a comprehensive analysis of how different automatic metrics align with human evaluation and find that MLLM-based metrics using GPT-4o are more reliable than Qwen-2.5 and traditional metrics such as DINO and CLIP.
>
> Our GPT-4o-based metric achieves a correlation of 0.7 with human judgments, which can be considered highly reliable and well-aligned. For comparison, widely adopted automatic metrics in the community show lower correlations: VIEScore [1] achieves a correlation of 0.4, while human-to-human correlation is only 0.45, and VQAScore [2] reaches 0.64, representing the state of the art among recent metrics.
>
> We acknowledge that there is still room to improve MLLM-based metrics and leave this for future work.
>
> ---
>
> ### **Q4. Robust and open automatic metrics**
>
> We agree that relying on a proprietary model may affect reproducibility, as API-based models can change over time. To address this, we have included the open-source model Qwen-2.5 as a judge in our evaluation. We are also extending our experiments to incorporate more advanced open-source MLLMs, such as Qwen-3 (released after the time of submission), and will present these results in the revised version.
>
> While the limitation of proprietary models exists, their use is common in widely adopted benchmarks because they often achieve higher correlation with human evaluation than open-source alternatives. For example, the image editing benchmarks ImgEdit [3] and GEdit [4] employ GPT-based judges.
>
> A key contribution of KITTEN is that it introduces the task of evaluating world entity generation and provides a structured benchmark, including a set of entities, prompts, and evaluation images. The focus is not on developing an MLLM judge; the benchmark can incorporate more powerful MLLMs over time, whether open-source or proprietary. We show that GPT-4o achieves a high correlation with human evaluation, but future MLLMs can be adopted to further improve alignment with human judgment, following common community practices for evolving and maintaining benchmarks.

---

> ### Author Response · Authors · 2025-12-12
>
> ### **Q5. Reliability analysis of human evaluation**
>
> The details are provided in Section J of the supplementary materials. We employed five annotators per image to ensure robust assessments. The raters are hired through Prolific.com, a third party rating service.
>
> For the binary task ("Adherence to Prompt Beyond References"), we observed agreement among at least 4 out of 5 annotators in 75% of cases and perfect agreement (5 out of 5) in 44% of cases.
>
> For the Likert scale task ("Faithfulness to Reference Entity"), we calculated Krippendorff’s Alpha at 0.60, indicating a good agreement for a subjective task of this complexity. Additionally, we achieved an IoU-like score of 0.53, which penalizes outliers and demonstrates moderate consensus, and an average pairwise Cohen’s Kappa of 0.25, reflecting fair pairwise agreement. The average standard deviation of ratings was 0.74, reflecting moderate variability in annotator judgments.
>
> These metrics collectively demonstrate the reliability of our human evaluation.
>
> ---
>
> ### **References**
>
> [1] Ku et al. *VIEScore: Towards Explainable Metrics for Conditional Image Synthesis Evaluation.* ACL, 2024.
>
> [2] Lin et al. *Evaluating Text-to-Visual Generation with Image-to-Text Generation.* ECCV, 2024.
>
> [3] Ye et al. *ImgEdit: A Unified Image Editing Dataset and Benchmark.* arXiv:2505.20275, 2025.
>
> [4] Liu et al. *Step1X-Edit: A Practical Framework for General Image Editing.* arXiv:2504.17761, 2025.

---

### Review · Reviewer_nWuG · 2025-11-25

**Summary Of Contributions:**

The authors propose KITTEN, a benchmark dataset and evaluation suite designed to assess the fine-grained "world knowledge" of text-to-image (T2I) generation models. While prior benchmarks focus on general text alignment or spatial reasoning (Section 2), KITTEN specifically targets the visual fidelity of real-world entities sourced from Wikipedia. The dataset spans 8 domains, covering both living entities like Insects and Plants as well as non-living ones like Landmarks and Vehicles (Fig. 2). It comprises approximately 6,440 prompts across 5 task categories such as "Composition" and "Style" (Section 3.2).
The paper conducts a systematic evaluation of three model families: Standard Backbone models like Flux and Imagen-3, Retrieval-Augmented models like DreamBooth, and Unified Understanding models (Section 4). To validate their findings, the authors establish a human evaluation protocol measuring "Faithfulness to Entity" and "Instruction Following" (Section 3.3) and demonstrate a strong correlation between these human scores and MLLM-based judges like GPT-4o (Table 3; Section 5.4).
Key Findings:
1. Empirical results show that while retrieval-augmented models significantly improve entity fidelity compared to backbone models, they suffer from reduced instruction-following capabilities. They often ignore style or composition constraints due to an over-reliance on the provided reference images (Section 4.2; Fig. 4).
2. Even state-of-the-art backbone models frequently hallucinate the visual details of less common entities, despite achieving high general text-alignment scores (Section 5.2).
3. The study demonstrates that standard text-alignment metrics (e.g., CLIP-T) are unreliable for assessing world knowledge. These metrics often assign high scores to backbone models that generate a generic version of the requested object (e.g., a generic rock formation) while failing to render the specific named entity (e.g., "Teufelsmauer") correctly (Section 5.2).
Strengths:
* The authors confirm the reliability of their MLLM-based judge by reporting strong correlations with human ground truth (Table 3), ensuring that the subsequent automated ablation studies reflect actual human perception rather than relying solely on noisy metrics like CLIP.
* The quantification of the trade-off between remembering what an object looks like versus following prompt instructions (Fig. 4 Left) provides a valuable insight for the future development of RAG-based image models.
Weaknesses:
* The dataset groups unique instances (e.g., specific Landmarks) with generic classes (e.g., Cuisine/Sports) under the single label of "Entity." Differentiating between these types would clarify the domain-specific analysis in Section 5.5, as the utility of retrieval likely differs when generating a unique object versus a broad category.
* Despite claiming to cover "real-world entities," the dataset excludes specific human identities (e.g., historical figures) found in related benchmarks like HEIM [1]. This limits the scope of "world knowledge" evaluation, as it omits a major category of real-world entities.

[1] Lee et al., “Holistic Evaluation of Text-to-Image Models,” NeurIPS, 2023.

**Additional Comments:**

None

**Audience:**

Yes

**Audience Explanation:**

As text-to-image models continue to scale and produce increasingly photorealistic outputs, assessing their factual reliability has become significantly more challenging. Recent research [2], highlights that traditional text alignment metrics like CLIP often fail to distinguish between generic representations and fine-grained, factually correct details. This submission aligns with that perspective by demonstrating that text descriptions alone are insufficient to convey the nuances of real-world visual knowledge, necessitating direct fidelity evaluation. Furthermore, the paper provides great insights for researchers in retrieval-augmented generation by rigorously demonstrating the trade-off between maintaining high entity fidelity and adhering to creative prompt instructions. These findings regarding the limits of current metrics and the behavior of RAG systems make the work highly relevant to the TMLR audience.

[2] Lin et al., 2024 refers to: Zhiqiu Lin, et al. "VQAScore: Evaluating Text-to-Image Generation with Visual Question Answering." ECCV, 2024.

**Broader Impact Concerns:**

No ethical concerns. The work focuses on evaluation and benchmarking, which inherently contributes to the safety and reliability of generative AI by quantifying hallucinations. However, since the objective is to improve the fidelity of "Real-World Entities," there is a theoretical dual-use risk regarding the creation of convincing misinformation (e.g., placing real landmarks or cultural artifacts in fabricated contexts). A brief acknowledgment of this potential would be a positive addition.

**Claims And Evidence:**

Yes

**Claims Explanation:**

A key strength of the submission is the reliance on human evaluation as the primary ground truth rather than solely relying on automated metrics. By implementing a strict protocol with distinct raters for "Faithfulness to Entity" and "Instruction Following" as detailed in Section 3.3, the authors effectively isolate the trade-off between visual accuracy and creative adherence. Furthermore, the paper provides clear statistical justification for using MLLMs to scale this evaluation. The reporting of strong Pearson and Spearman correlations of approximately 0.7 between human judgments and the GPT-4o judge in Table 3 ensures that the subsequent large-scale ablation studies are grounded in reality rather than noisy metrics.
Additionally, the quantitative findings are convincing because they converge with the qualitative evidence. The aggregate data showing a drop in instruction-following scores for retrieval models in Figure 4 is supported by clear visual failure cases in Figures 6 and 7, such as retrieval models ignoring composition prompts to preserve entity structure. This alignment between the statistical graphs and the visual samples makes the findings regarding the fidelity-creativity trade-off highly persuasive. The authors also successfully demonstrate the inadequacy of existing metrics by showing that high CLIP-T scores often mask visual hallucinations in backbone models, as seen in the analysis in Section 5.2.
The evidence provided is accurate for the tested domains. However, I noted the scope in the summary because the empirical results focus on non-human objects and species. Extending the evaluation to include specific human identities would further support the generalizability of the findings across all real-world entities. Additionally, the analysis in Section 5.5 combines unique instances (e.g., specific landmarks) with generic classes (e.g., cuisine) into single domain scores. Distinguishing between these types would help clarify whether low scores are due to challenges in retrieving a unique instance or generating a generic category.

**Requested Changes:**

Requested Changes

1. Clarification of "Entity" Scope (Strengthening): The paper defines "Visual Entities" across 8 domains (Fig. 2). While this includes living things (Insects, Plants), it appears to explicitly exclude specific human identities (e.g., celebrities, historical figures), unlike related benchmarks such as HEIM [1].
* Request: Please add a brief justification for the exclusion of specific human identities in the introduction or discussion. Clarifying whether this is due to ethical constraints (privacy) or technical constraints (different metrics required for face ID) would help readers better understand the specific scope of "world knowledge" being evaluated.
2. Distinction between Instances and Classes in Section 5.5 (Strengthening): In Section 5.5, the analysis groups unique instances (e.g., specific Landmarks like "Bandinelli Palace") with generic classes (e.g., Cuisine like "Guacamole" or Sports like "Snowboarding").
* Request: Please include a brief discussion on how this distinction impacts the performance of retrieval-augmented models. It is intuitive that retrieval mechanisms serve a different function when the target is a unique object not present in the base training data versus a generic category that the model likely understands. Acknowledging this semantic difference would add depth to the domain-specific analysis.

---

> ### Author Response · Authors · 2025-12-13
>
> ### **Q1. Justification for the exclusion of specific human identities**
>
> We intentionally exclude specific human identities from our dataset due to ethical and privacy considerations. Our institution enforces strict guidelines requiring explicit consent from individuals before using any images containing their likeness. To respect privacy and comply with these rules, our dataset focuses exclusively on non-human objects and species.
>
> Our domains are sampled from OVEN-Wiki, which is constructed from entities in 14 existing image recognition and visual question answering datasets, where all entity labels are grounded to Wikipedia. We sample eight representative domains from the image recognition portions of OVEN-Wiki, including iNaturalist2017, Cars196, Food101, Sports100, Aircraft, Oxford Flowers, and Google Landmarks v2, which correspond to the plant, insect, vehicle, cuisine, sport, aircraft, flower, and landmark domains in KITTEN.
>
> ---
>
> ### **Q2. Distinction between instances and classes in Section 5.5**
>
> We acknowledge that the eight domains in our dataset include both instance-level categories (e.g., landmarks) and class-level categories (e.g., cuisine and sports). To clarify these distinctions, we categorize them along a spectrum of visual variability:
>
> 1. Generic classes, such as sports or cuisine, exhibit the highest inter-category variation, as items within the same category can differ substantially in appearance.
> 2. Specific classes, such as insects and flowers, show moderate variation, since different items within the same species can still look visually distinct.
> 3. Generic instances, such as aircraft or vehicles, have lower variation, as different instances of the same model tend to share similar structures.
> 4. Specific instances, such as landmarks, exhibit the lowest variation because each has a single, fixed visual appearance.
>
> Here, we continue the domain-specific analysis from Section 5.5 and clarify how domain distinctions affect the performance of retrieval-augmented models.
>
> For generic classes (e.g., cuisine, sports), the retrieved images exhibit higher variation and are less representative. As a result, the outputs of retrieval-augmented models tend to resemble the target images less closely, leading to lower performance compared to the backbone models. In contrast, for specific instances (e.g., landmarks), the retrieved images show lower variation and are highly representative. This allows the outputs of retrieval models to be consistent and closely aligned with the target images, resulting in higher scores than the backbone models.
>
> From another perspective, the domains can be categorized by their frequency in the training data:
>
> 1. Generic classes, such as sports or cuisine, appear most frequently.
> 2. Generic instances, such as aircraft or vehicles, and specific classes, such as insects, plants, and flowers, occur with moderate frequency.
> 3. Specific instances, such as landmarks, appear least frequently.
>
> Consequently, retrieval-augmented models outperform backbone models in specific domains (e.g., landmarks), where the target entities are rarely seen in the training data and are underrepresented in the backbone model’s parameters. Retrieval-augmented models enhance performance by incorporating reference images during inference. In contrast, retrieval-augmented models underperform compared to backbone models in generic domains (e.g., cuisine, sports), where the concepts are common and already well-memorized by the backbone model. In these cases, retrieval models may perform worse due to overfitting on a limited set of reference images.
>
> We also note that the reference images used in our experiments are not obtained from a real-world retrieval pipeline. Instead, they are sampled directly from OVEN-Wiki and therefore represent highly accurate matches with a very low error rate across all domains. As a result, we do not attribute the observed performance differences to limitations of the retrieval engine.

---

### Review · Reviewer_3Fq9 · 2025-11-27

**Summary Of Contributions:**

The paper introduces KITTEN, a benchmark and evaluation suite for knowledge intensive text to image generation that focuses on visual entity fidelity rather than only aesthetic quality or text prompt alignment. It constructs 6,440 prompts over 322 Wikipedia entities spanning eight domains, provides reference images and support sets, and defines five task types (basic, location, composition, style, material), with a human evaluation protocol that separately measures entity faithfulness and instruction following along with automatic metrics (CLIP, DINO) and MLLM based judges. Using this benchmark, the authors systematically evaluate backbone, retrieval augmented, and unified understanding generation models, and empirically characterize tradeoffs between entity fidelity and instruction following, as well as the limitations of existing automatic metrics. Strengths include a clear problem formulation (visual world knowledge in generation), reasonably broad coverage of entities, a thoughtfully designed human evaluation pipeline, and extensive comparative results that highlight non obvious behavior of retrieval augmentation and unified models. Main weaknesses are that the entities and prompts, while diverse, are still limited to Wikipedia and eight domains, some important dataset and annotation details are relatively high level, and the benchmark currently focuses on single entity fidelity and simple compositionality rather than more complex reasoning or multi entity scenes.

**Audience:**

Yes

**Audience Explanation:**

TMLR’s audience includes researchers working on generative models, evaluation methodologies, and multimodal reasoning, all of whom would likely benefit from a benchmark that specifically targets factual visual entity knowledge in text to image models, a dimension that is increasingly important as these systems are deployed in user facing tools. The work provides a structured way to probe what current models actually know about real world entities visually, and it distinguishes between different families of approaches (plain backbones, retrieval augmented methods, unified models) in a way that informs both algorithm design and system engineering. The analysis of automatic metrics versus human ratings and MLLM judges is also relevant to a broader community that is grappling with how to evaluate generative systems at scale. Even if some readers do not directly use KITTEN, the design choices, observed tradeoffs between entity fidelity and instruction following, and domain specific performance differences provide conceptual insights about text to image model behavior that are of general interest to TMLR readers.

**Broader Impact Concerns:**

The work touches on several broader impact issues that could be discussed more explicitly. Because KITTEN focuses on real world entities with reference images from Wikipedia, there are potential concerns around copyright, privacy, and representational bias in the underlying images (for example, geographic or cultural skew in which landmarks, cuisines, sports, or plants are covered and how they are depicted), and it would be helpful to spell out how these risks were assessed and mitigated and what usage constraints apply to the benchmark. There is also the possibility that a benchmark that emphasizes entity fidelity might incentivize model developers to memorize web imagery more aggressively, which could exacerbate copyright and privacy risks for individuals or locations that appear in training data. Finally, the benchmark and evaluation pipeline could be used to rank and market commercial models on “knowledgeable” image generation without adequately communicating limitations, which may mislead end users about factual reliability. I would therefore suggest adding or expanding a Broader Impact Statement that discusses dataset curation and licensing, potential misuse cases and mitigations, and the balance between encouraging better entity fidelity and avoiding harmful overfitting to web sourced images.

**Claims And Evidence:**

Yes

**Claims Explanation:**

The core claims are that (i) current text to image and unified models often fail to accurately reproduce fine grained visual details of real world entities, (ii) retrieval augmented methods substantially improve entity fidelity but often degrade instruction following and text alignment, and (iii) strong MLLMs can serve as reasonably correlated automatic judges for this benchmark. These are supported by a clearly described benchmark construction process from OVEN Wiki entities and Wikipedia images, a human evaluation protocol with two decomposed criteria and five raters per image, and quantitative results that consistently show retrieval augmented models increasing entity faithfulness while reducing instruction following, as well as backbone improvements that help both but do not fully close the entity fidelity gap. The paper further backs the MLLM evaluation claim with a correlation analysis comparing GPT 4o and Qwen 2.5 scores against human scores and traditional metrics such as CLIP T and DINO, showing higher correlations for MLLMs. Remaining concerns are mostly about scope and external validity rather than internal support: the benchmark is limited to the chosen domains and knowledge source, there is little ablation on prompt generation and annotation instructions, and some results rely on proprietary systems whose training data and interfaces are not fully controlled, but none of these issues obviously undermines the main empirical conclusions.

**Requested Changes:**

I would recommend the following adjustments: a) provide more detailed statistics and documentation about the dataset, including how entities and images were filtered from OVEN Wiki and Wikipedia, how many images per entity are in the support versus evaluation sets, and any licensing or quality filtering applied; b) expand the description of the human evaluation protocol, including annotator pool (expertise, geography), pay, quality control measures, and inter annotator agreement, since human judgment is central to the paper’s claims; c) clarify the prompt generation process from ChatGPT with concrete examples and error checks, and discuss how prompt wording might bias results; d) make the evaluation setup for each model family more explicit, especially for proprietary or API based systems (for example, prompt formats, temperature, number of samples, selection of images), and provide enough detail that others can reproduce the experiments; e) strengthen the discussion of limitations and external validity, for example with respect to the choice of Wikipedia entities, single entity focus, and lack of reasoning heavy prompts; f) include more analysis at the level of specific failure modes, for example qualitative error taxonomies per model family (wrong global shape, wrong material, spurious attributes) and a breakdown by entity frequency or popularity; g) release a detailed datasheet or documentation with scripts and annotation templates linked from the camera ready version.

---

> ### Author Response · Authors · 2025-12-13
>
> ### **Q1. Detailed statistics about the dataset**
> Our domains are sampled from OVEN-Wiki, which is constructed from entities in 14 existing image recognition and visual question answering datasets, where all entity labels are grounded to Wikipedia. We sample eight representative domains from the image recognition portions of OVEN-Wiki, including iNaturalist2017, Cars196, Food101, Sports100, Aircraft, Oxford Flowers, and Google Landmarks v2, which correspond to the plant, insect, vehicle, cuisine, sport, aircraft, flower, and landmark domains in KITTEN.
>
> For each domain, we select up to 50 entities. If a domain contains more than 50 entities, we first exclude entities with high Wikipedia page-click counts and then randomly sample 50 from the remaining set. This results in 322 entities across eight domains. For each entity, we randomly sample 10 images from OVEN-Wiki as the support set and 5 images as the evaluation set. The detailed statistics are provided in Section I of the supplementary materials.
>
> Quality filtering uses a general safety filter to remove non-imageable classes, classes containing undesired social bias, and non-entity classes. Our dataset will be released under the Apache-2.0 license, consistent with OVEN-Wiki.
>
> ---
>
> ### **Q2. Description of the human evaluation protocol**
> We employed five annotators per image to ensure robust assessments. The raters are hired through Prolific.com, a third-party rating service. For the binary task "Adherence to Prompt Beyond References," we observed agreement among at least 4 out of 5 annotators in 75% of cases and perfect agreement (5 out of 5) in 44% of cases. For the Likert scale task "Faithfulness to Reference Entity," we calculated Krippendorff’s Alpha at 0.60, indicating a good agreement for a subjective task of this complexity. Additionally, we achieved an IoU-like score of 0.53, which penalizes outliers and demonstrates moderate consensus, and an average pairwise Cohen’s Kappa of 0.25, reflecting fair pairwise agreement. The average standard deviation of ratings was 0.74, reflecting moderate variability in annotator judgments. These metrics collectively demonstrate the reliability of our human evaluation. Human Annotation Instructions are included in Section L of the supplementary materials.

---

> ### Author Response · Authors · 2025-12-13
>
> ### **Q3. Prompt generation process from ChatGPT**
> We use ChatGPT to generate text-to-image prompts for each domain according to five evaluation tasks: Basic, Location, Composition, Style, and Material. Each prompt explicitly mentions the entity’s domain. The authors then conduct an error check by manually filtering and revising the prompts as needed. An example is provided below.
> ```
> # Instructions to ChatGPT
> You are tasked with generating text-to-image prompts for an evaluation benchmark. The goal is to create prompts for five specific evaluation tasks. For each entity provided, generate one prompt per task. Each prompt should explicitly include the entity’s domain while avoiding the entity’s exact name; for example, use the <sks> <domain> instead of the entity’s name. Below are the evaluation tasks with example templates (for reference). The outputs should follow the same structure but use the flower domain.
>
> Evaluation Tasks
> 1. Basic Prompt
> Generate a simple description of the entity with its domain.
> Prompt: Photo of <sks> landmark.
> 2. Entity in a Specified Location
> Place the entity in a particular environment or location, explicitly including its domain.
> Prompt: <sks> landmark stands against snow-capped mountains.
> 3. Composition with Other Objects
> Combine the entity with other objects in a meaningful composition, explicitly including the entity’s domain.
> Prompt: A peacock in front of <sks> landmark.
> 4. Entity in Specific Styles
> Depict the entity in a particular artistic style, explicitly including its domain.
> Prompt: An oil painting of <sks> landmark.
> 5. Entity Made of Specific Materials
> Represent the entity as if it is made from various materials, explicitly including its domain.
> Prompt: The <sks> landmark made of crystal.
>
> # Outputs from ChatGPT
> 1. Basic Prompt
> Prompt: Photo of the <sks> flower.
> 2. Entity in a Specified Location
> Prompt: The <sks> flower blooms in a sunlit meadow.
> 3. Composition with Other Objects
> Prompt: A butterfly rests on the <sks> flower.
> 4. Entity in Specific Styles
> Prompt: A watercolor painting of the <sks> flower.
> 5. Entity Made of Specific Materials
> Prompt: The <sks> flower sculpted from glass.
> ```
>
> Prompt wording can bias evaluation results, as even small changes in phrasing may influence how models interpret the entity. For example, adding stylistic adjectives or specifying a particular viewpoint could unintentionally advantage or disadvantage certain models. KITTEN, however, focuses on assessing a model’s ability to reconstruct the visual entity under straightforward modifications, rather than testing language understanding or reasoning required for complex prompts beyond the main entity. Therefore, we use simple and direct prompts (average length 9.91 words, standard deviation 2.86) across the five evaluation tasks, minimizing the impact of prompt wording variations.
>
> ---
>
> ### **Q4. Evaluation setup for each model family**
> To facilitate reproducibility, we provide detailed descriptions of the inference setups for all evaluated models. All models generate a single sample for each prompt without any selection or reranking.
>
> For backbone text-to-image models (DALL·E-2, SD-1.5, SD-2.1, SD-3, SD-XL, Imagen, Flux, Imagen-3) and unified models (Show-o, Janus-Pro, Emu3), we follow their released inference configurations, where no temperature parameter is used. These models do not require fine-tuning and do not take support images as inputs; they directly generate an image from the input prompt.
>
> For retrieval-augmented models, the in-context learning methods (BLIP-Diffusion, IP-Adapter, Instruct-Imagen) also do not require fine-tuning. They take a single support image together with the text prompt as input; we randomly select the support image. For fine-tuning-based methods (Custom-Diff, DreamBooth), we fine-tune the model for each entity using 10 support images. We use the AdamW optimizer with a learning rate of 5×10⁻⁶ and default beta values (β₁ = 0.9, β₂ = 0.999). Training is performed for 1,000 steps with a batch size of 5. After fine-tuning, these models take only the text prompt as input to generate an image.
>
> Evaluation with GPT-4o-mini and Qwen-2.5-VL-7B is conducted via API calls with temperature set to 0. The prompt format is provided in Section M of the supplementary materials. The MLLM is asked to answer two questions separately:
>
> 1. How faithfully the generated image represents the entity mentioned in the prompt, given the prompt, five reference images displayed in a row, and the generated image.
> 2. How well the generated image reflects the details described in the prompt, given only the prompt and the generated image.

---

> ### Author Response · Authors · 2025-12-13
>
> ### **Q5. Discussion of limitations and external validity**
> We discuss the limitations of multi-entity prompts, complex reasoning prompts, and choices of Wikipedia entities below.
>
> **Multi-entity prompts.**
> KITTEN focuses on single-entity prompts because our goal is to stress-test a model’s ability to reconstruct visual entities from diverse domains under a basic, controlled setup. Multi-entity prompts pose additional challenges, as the model must preserve each entity’s identity without mixing or confusion, a task that is not unique to specialized entities and can be equally difficult for common objects. Although multi-entity generation is more complex, retrieval-augmented models such as Custom Diffusion have shown that, under the same fine-tuning procedure, a model can handle multiple target entities with quality comparable to single-entity generation. Therefore, we believe that our findings can be extended to multi-entity settings. In particular, the trade-off we observe between maintaining high entity fidelity and following creative prompt instructions is likely to persist. Our proposed future direction, combining a strong backbone with a well-designed retrieval-augmented approach, should also hold in multi-entity scenarios.
>
> **Complex reasoning prompts.**
> KITTEN does not include prompts that require complex reasoning. We observe that several concurrent benchmarks evaluate models using complex prompts involving implicit reasoning (e.g., WISE, WorldGenBench, R2I-Bench, OmniGenBench, ABP), but none focus on generating a broad set of real-world entities across diverse domains, which is the primary focus of KITTEN. As noted in the WISE paper, many complex reasoning prompts (e.g., “the tallest building in Manhattan”) can be reformulated into entity-centered prompts through prompt rewriting, which consistently improves model scores. Consistent with this observation, our analysis intentionally abstracts away the “language reasoning” component and instead focuses on the core challenge of “visual entity generation” using explicitly instructed prompts that effectively represent the rewritten versions.
>
> **Choices of Wikipedia entities.**
> While KITTEN prompts are currently limited to Wikipedia and eight domains, it covers the largest number of visual entities among existing benchmarks, including 322 entities across 8 domains and 6,440 prompts. KITTEN provides broader coverage than alternative image customization benchmarks, which focus on different tasks and include fewer entities and prompts. For example, DreamBench [1], DreamBench-v2 [2], CustomConcept101 [3], DreamBench++ [4], MM-Diff [5], and StoryMaker [6] evaluate only 30, 30, 101, 150, 25, and 40 subjects, respectively, with 750, 220, 2020, 1350, 500, and 800 prompts.
>
> We emphasize that this work represents an initial step toward building a large-scale evaluation dataset for the novel task of generating real-world visual entities, and we plan to expand its scope in future work. This includes incorporating domains not yet covered, such as the VQA data in OVEN-Wiki, as well as extending the dataset by selecting visually relevant categories and entities from resources like OntoNotes and WordNet.
>
> ---
>
> ### **References**
> [1] Ruiz et al. *DreamBooth: Fine-tuning text-to-image diffusion models for subject-driven generation.* CVPR, 2023.
>
> [2] Chen et al. *Subject-driven Text-to-Image Generation via Apprenticeship Learning.* NeurIPS, 2023.
>
> [3] Kumari et al. *Multi-concept Customization of Text-to-Image Diffusion.* CVPR, 2023.
>
> [4] Peng et al. *DreamBench++: A Human-Aligned Benchmark for Personalized Image Generation.* arXiv:2406.16855, 2024.
>
> [5] Wei et al. *MM-Diff: High-Fidelity Image Personalization via Multi-Modal Condition Integration.* arXiv:2403.15059, 2024.
>
> [6] Zhou et al. *StoryMaker: Towards Consistent Characters in Text-to-Image Generation.* arXiv:2409.12576, 2024.

---

> > ### Author Response · Authors · 2025-12-13
> >
> > ### **Q6. Analysis and specific failure modes**
> > Unified models show strong instruction-following ability but perform noticeably worse in maintaining entity fidelity. For low-frequency categories, such as plants, insects, and landmarks, they often produce entities that bear little or no resemblance to the target. In some cases, they even generate completely unrelated content, such as a person, when prompted with rare landmarks or insects. Although high-frequency entities are more reliably produced, the models still struggle with common items like Peking duck and macaron, revealing persistent challenges in accurate entity generation.
> >
> > Retrieval-based models that rely on in-context learning, including BLIP-Diffusion, IP-Adapter, and Instruct-Imagen, tend to over-rely on the reference image and sometimes nearly copy it into the output. These models can preserve high entity fidelity, but their ability to follow instructions is limited. They rarely generate new materials or styles for the target entity and often fail to incorporate additional objects alongside the main subject.
> >
> > Retrieval-based models that rely on fine-tuning, such as Custom-Diff and DreamBooth, achieve a better balance between instruction following and entity fidelity, although they still fall short of the strongest backbone models in instruction following. They produce good fidelity for low-frequency categories, like plants and landmarks, but fine-tuning instability occasionally causes distortions in the entity’s shape or structure. This instability also leads to weaker performance on high-frequency entities, such as cuisine. In compositional prompts involving additional objects, these models often struggle because they may omit the main entity or the secondary objects. They also have difficulty rendering the entity in alternative materials.
> >
> > Turning to backbone models, the SD family shows strong overall performance. Both SD3 and SDXL exhibit strong instruction-following ability and high entity fidelity, with SDXL performing even better. These models handle high-frequency entities, such as cars and cuisine, well, but struggle with low-frequency entities, like plants and landmarks. In these cases, the generated examples often fail to match the target’s attributes or deviate significantly in structure and configuration. The Imagen family shows similar trends. Imagen-3 demonstrates exceptionally strong instruction following and high entity fidelity, performing well on high-frequency entities while still facing challenges with low-frequency ones. A notable strength of Imagen-3 is its strong compositional capability. It successfully generates challenging scenarios, such as an entity wearing sunglasses or floating in a pool, which most other models fail to produce accurately.
> >
> > ---
> >
> > ### **Q7. Dataset release**
> > We will release our datasets and documentation in the camera-ready version, including entities, prompts, and images. The release will also include the evaluation scripts for both automatic metrics and MLLM-based judging.

---

### Decision · Action_Editor_eXt3 · 2025-12-24

**Recommendation:** Accept with minor revision

**Additional Comments:**

Please ensure the camera-ready includes:

Tightened claims and terminology: Replace or carefully qualify “knowledge-intensive” framing to avoid implying reasoning-based evaluation when the benchmark is explicitly entity-centered.

Instance vs class analysis surfaced in the main text: The authors’ spectrum-based categorization is helpful — please ensure it is integrated into the main discussion of domain effects so readers understand why retrieval helps in some domains and can hurt in others.

Dataset documentation completeness: Include the promised dataset statistics, selection/filtering criteria, licensing notes, and annotation instructions in a durable, easy-to-find form (e.g., a datasheet), plus links in the main paper.

**Audience:**

Yes

**Audience Explanation:**

Across reviewers, there is consistent agreement that the audience would care about the findings because the work targets a concrete and increasingly important evaluation gap: whether text-to-image systems can faithfully render specific real-world visual entities, not merely produce aesthetically pleasing or loosely aligned images. The benchmark resource, the human-centered protocol, and the systematic comparison across backbone, retrieval-augmented, and unified models yield broadly useful takeaways. The analysis that common automatic metrics can miss entity-level errors, is also directly relevant to evaluation research in generative modeling.

**Claims And Evidence:**

Yes

**Claims Explanation:**

The core claims are supported by a well-specified benchmark design plus a human-evaluation methodology. Two reviewers explicitly judge the evidence as convincing and clear, highlighting that the benchmark isolates entity faithfulness vs instruction following with multiple annotators and that the paper substantiates the inadequacy of common automatic alignment metrics for entity-level correctness. One reviewer initially marked “No,” but their blocking concerns were addressed in the author responses: the authors added inter-rater reliability and variability statistics, and they clarified prompt generation and evaluation setups.